# *Mycobacterium tuberculosis* overcomes phosphate starvation by extensively remodelling its lipidome with phosphorus-free lipids

Robert M. Gray [1,2,3] ✉, Debbie M. Hunt[1,8], Mariana Silva dos Santos [1,4], Jiuyu Liu [5], Aleksandra Agapova[1], Angela Rodgers[6], Antony Fearns[6], Julio Ortiz Canseco[6], Acely Garza-Garcia[1], James I. MacRae [4], Maximiliano G. Gutierrez [6], Richard E. Lee [5] & Luiz Pedro S. de Carvalho [1,7] ✉

Tuberculosis (TB) is the biggest cause of death from infectious disease worldwide. The causative agent, *Mycobacterium tuberculosis* (Mtb), possesses a complex cell envelope comprised of multiple classes of unique lipids. The macrophage phagosome is a key reservoir of infection in pulmonary TB and multiple studies have shown that inorganic phosphate (Pi) is limiting in this environment. Here, we show that during Pi restriction the Mtb lipidome markedly remodels such that phospholipids are replaced with multiple classes of phosphorus-free lipids. This envelope lipidome remodelling suggests that standard Mtb culture conditions that use media with high concentrations of Pi do not reflect the physiologic environment during infection, thereby undermining vaccine and drug development for tuberculosis. Further, we discover that Mtb can metabolise phospholipid polar heads abundant in host pulmonary surfactant as an alternative phosphate source. Therefore, we present two mechanisms where Mtb manipulates lipid metabolism to overcome host restriction.

Inorganic phosphate (Pi) is the preferred source of phosphorous for bacteria, an essential component of a plethora of biomolecules. Studies have shown that the Pi starvation response system in Mtb, orchestrated by the two-component system, comprising sensor histidine kinase SenX$_3$ and transcriptional response regulator RegX$_3$, are required for full virulence in animal models of TB and are essential for Mtb growth within human macrophages in vitro[1–4]. There is additional evidence from multiple modes of experiment that show that the macrophage phagosome is a Pi restricted compartment:

*Mycobacterium marinum* engineered to fluoresce when encountering a Pi concentration of <10 μM emits fluorescence from 3 days post infection of zebrafish embryos, an extensively used model of TB, and a similarly sensitive reporter in *Salmonella enterica* fluoresces during infection of macrophages but not during infection of epithelial cells, suggesting that Pi starvation is specific to the macrophage phagosome[5,6].

Environmental and pathogenic bacteria, and some yeasts, have been shown to be able to substitute phospholipids for phosphorus-

[1]Mycobacterial Metabolism and Antibiotic Research Laboratory, The Francis Crick Institute, London, UK. [2]Division of Infection & Immunity, UCL, London, UK. [3]Department of Chemistry, Imperial College London, London, UK. [4]Metabolomics Scientific Technology Platform, The Francis Crick Institute, London, UK. [5]Department of Chemical Biology & Therapeutics, St. Jude Children's Research Hospital, Memphis, TN, USA. [6]Host-Pathogen Interactions in Tuberculosis Laboratory, The Francis Crick Institute, London, UK. [7]Department of Chemistry, The Herbert Wertheim UF Scripps Institute for Biomedical Innovation and Technology, Jupiter, FL, USA. [8]Deceased: Debbie M. Hunt. ✉e-mail: rekgrgr@ucl.ac.uk; soriodecarval.lp@ufl.edu

free lipid species when Pi is limiting[7–14], a strategy also common in plants and green algae[15–17]. Mtb, however, contains a particularly complex lipidome, organized into a multi-layered envelope composed of an inner plasma membrane (PM) and an outer mycomembrane (MOM) (Supplementary Fig. 1a–c), and this envelope is a key physiological and virulence determinant of Mtb. Whilst the MOM is composed of highly hydrophobic, complex lipids, many found only in mycobacteria and the closest relatives, the PM is composed principally of phospholipids, and their glycosylated derivatives phosphatidylinositol mannosides (PIMs).

The enzymatic pathway for the breakdown of phospholipids in Mtb has been proposed to involve the sequential action of yet to be identified phospholipases and a glycerophosphodiesterase, followed by the action of the *rv1692*-encoded glycerol-3 phosphate phosphatase (Supplementary Fig. 1b). We previously characterized the action of Rv1692[18], which produces glycerol and phosphate as the final products of the pathway. Mtb is known to actively transport exogenous phospholipid polar heads across its PM via the ATP-binding cassette transporter UgpABCE[19,20], and intriguingly, studies have indicated that this transporter is transcriptionally induced by RegX₃[1,21], suggesting a link between polar head uptake and phosphate starvation. It is also known that Mtb remodels its envelope composition in response to external stimuli such as low oxygen tension or physiologic salinity[22–24]. Here, in order to probe the ability of Mtb to adapt its phospholipid processing to low phosphate conditions, we study the proposed glycerophosphodiesterase in the pathway, GlpQ1, and we discover a dramatic change in the Mtb envelope lipid composition in response to phosphate restriction, involving >1500 lipid species; and also propose a role for GlpQ1 in the metabolism of host-derived phospholipid heads, ultimately providing two mechanisms for Mtb to overcome Pi limitation during human infection.

## Results

### *glpQ1* deletion disrupts phospholipid remodelling

GlpQ1, encoded at locus *rv3842c* in the virulent reference strain of Mtb; H37Rv, is annotated by sequence homology to be a glycerophosphodiesterase; an enzyme that hydrolyses the polar lipid-heads of glycerophospholipids following their deacylation by phospholipases (Supplementary Fig. 1b). To probe the enzymatic function of GlpQ1 and its role in PM remodeling we generated an in-frame clean genetic deletion of *glpQ*1 in H37Rv (Δ*glpQ*1 strain) and performed unbiased global metabolomics paired with lipidomics. Parent/wild type (WT), Δ*glpQ*1 and the complement strain Δ*glpQ*1::*glpQ*1_*Pimyc*, growing exponentially in liquid culture, were subjected to a modified Bligh-Dyer extraction[25], followed by biphasic partitioning. The aqueous phase of these extracts contains the cell's polar metabolites, which includes the four glycerophosphodiesters of Mtb, the proposed substrates for GlpQ1. Employing a liquid chromatography mass spectrometry (LCMS) protocol hereafter referred to as the HILIC method (hydrophilic interaction liquid chromatography), we analyzed 1064 detected features in positive ion-mode and 616 in negative ion-mode. Whilst >96% of features in the Δ*glpQ*1 strain were not significantly altered compared with WT, 3 of the 4 canonical polar-lipid heads of Mtb, glycerophosphoethanolamine (GroPEth), glycerophosphoglycerol (GroPGro) and glycerophosphoinositol (GroPIns) markedly enriched in the Δ*glpQ*1 strain, with fold changes of >64-fold ($2^6$) (Fig. 1a and Supplementary Fig. 2a). The 4th lipid head, bisglycerophosphoglycerol (Bis(GroP)Gro), was detected but the chromatographic resolution was poor. We therefore analyzed polar cell extracts using a second LCMS method (amide column), which confirmed Bis(GroP)Gro also accumulated markedly in Δ*glpQ*1. (Fig. 1b and Supplementary Fig. 2b). Consistent across both LCMS methods and all four lipid-heads, the phenotype showed partial complementation, with ~50% reversion of lipid-head levels to that of the WT in Δ*glpQ*1::*glpQ*1_*Pimyc* (Fig. 1b and Supplementary Fig. 2b). To address

the only partial reversion seen in this metabolic phenotype, we subsequently generated a second complement strain, with the *glpQ*1 gene now under the strong myc promotor[26] Δ*glpQ*1::*glpQ*1_*Psmyc*. In this strain, almost complete reversion to wild-type levels was seen for the polar heads (Supplementary Fig. 2b). These findings strongly support the physiological role of GlpQ1 as a polar head glycerophosphodiesterase in Mtb. In further support, a $2^7$ fold accumulation was seen in Δ*glpQ*1 in an ion presumptively identified as glycerophosphothreonine (GroPThr) (Fig. 1a and Supplementary Fig. 2a, c). Higher energy collisional dissociation mass spectrometry (HCD) clearly supported this identification and was consistent with collision spectra produced in studies of the corresponding acylated lipid, phosphatidylthreonine[27,28]. We conclude either that Mtb may also produce additional minor species of phospholipids containing amino acids, or may import phospholipid heads from the surrounding milieu, that are usually rapidly metabolized in a GlpQ1 dependent pathway, and so GroPThr is only seen when *glp*Q1 is deleted. Supplementary Data 1 details the ion features for all six lipid-heads detected, and Supplementary Notes 1 related HCD spectra.

By analyzing the global lipidome of the strains, we were able to investigate how the marked accumulation of the lipid-heads in the cytosolic compartment translated into PM changes. By using a mycobacterial lipidome platform[29] for analysis of the organic phase of the cell-extracts (Fig. 1c and Supplementary Fig. 3a–d), we detected 3629 lipid features in positive ion-mode, and of these 125 lipids (3.5%) were significantly altered in the Δ*glpQ*1 strain. Analysis of the changes in phospholipid levels in the Δ*glpQ*1 strain versus the WT, at the level of phospholipid class (Fig. 1c), revealed that phosphatidylglycerol (PG) and phosphatidylinositol (PI) levels both fell by ~40% each, and this was consistent across two independent experiments. Further, these changes showed near total complementation in the Δ*glpQ*1::*glpQ*1_*Pimyc* strain. By contrast, there were no robust changes in total levels of the other phospholipids, phosphatidylethanolamines (PEs), cardiolipins (CLs) or acylated phosphatidylinositol dimannosides (PIM₂s), despite the large accumulation of their corresponding lipid-heads in the cytosolic compartment.

PIM₂s are a major component of the Mtb PM, and the finding that their levels do not alter significantly between the strains (Fig. 1c and Supplementary Fig. 3c, d) is interesting, as PIMs are constructed on a PI membrane anchor, and therefore this conservation of PIM₂ levels in Δ*glpQ*1 suggests that available PI is prioritised for PIM₂ synthesis. Both PI and PIMs are essential in mycobacteria, with loss of viability of *Mycobacterium smegmatis* observed when levels decrease by 70% and 50%, respectively[30]. The findings that phospholipid levels in Δ*glpQ*1 moderately decrease in 2 of 5 classes and remain essentially constant in 3 of 5, despite all corresponding lipid head species markedly accumulating within the metabolome, indicates that robust homeostatic mechanisms must exist at the PM to maintain its composition and function in light of perturbations in the lipid-head pool. Supplementary Data 2 details the ion features for all phospholipids detected, Supplementary Notes 1 shows example HCD spectra for the same, and Supplementary Notes 2 shows the MS signals for each individual phospholipid species.

### GlpQ1 hydrolyses exogenous lipid heads to maintain growth

An unexpected finding in our untargeted metabolomic analysis was the detection of glycerophosphocholine (GroPCho), which was observed in positive ion-mode and was enriched in the Δ*glpQ*1 mutant similarly to other lipid-heads of Mtb. (Fig. 2a–c and Supplementary Fig. 2a). GroPCho is the lipid-head of phosphatidylcholine (PC), and a major phospholipid of mammalian cells that is also found in some bacteria, but not thought to be present in Mtb. An analysis of the growth media used in the experiments detected multiple PC species, both in the Albumin-Dextrose-Catalase (ADC) growth supplement and in the Middlebrook medium itself, reflecting the ubiquitous nature of

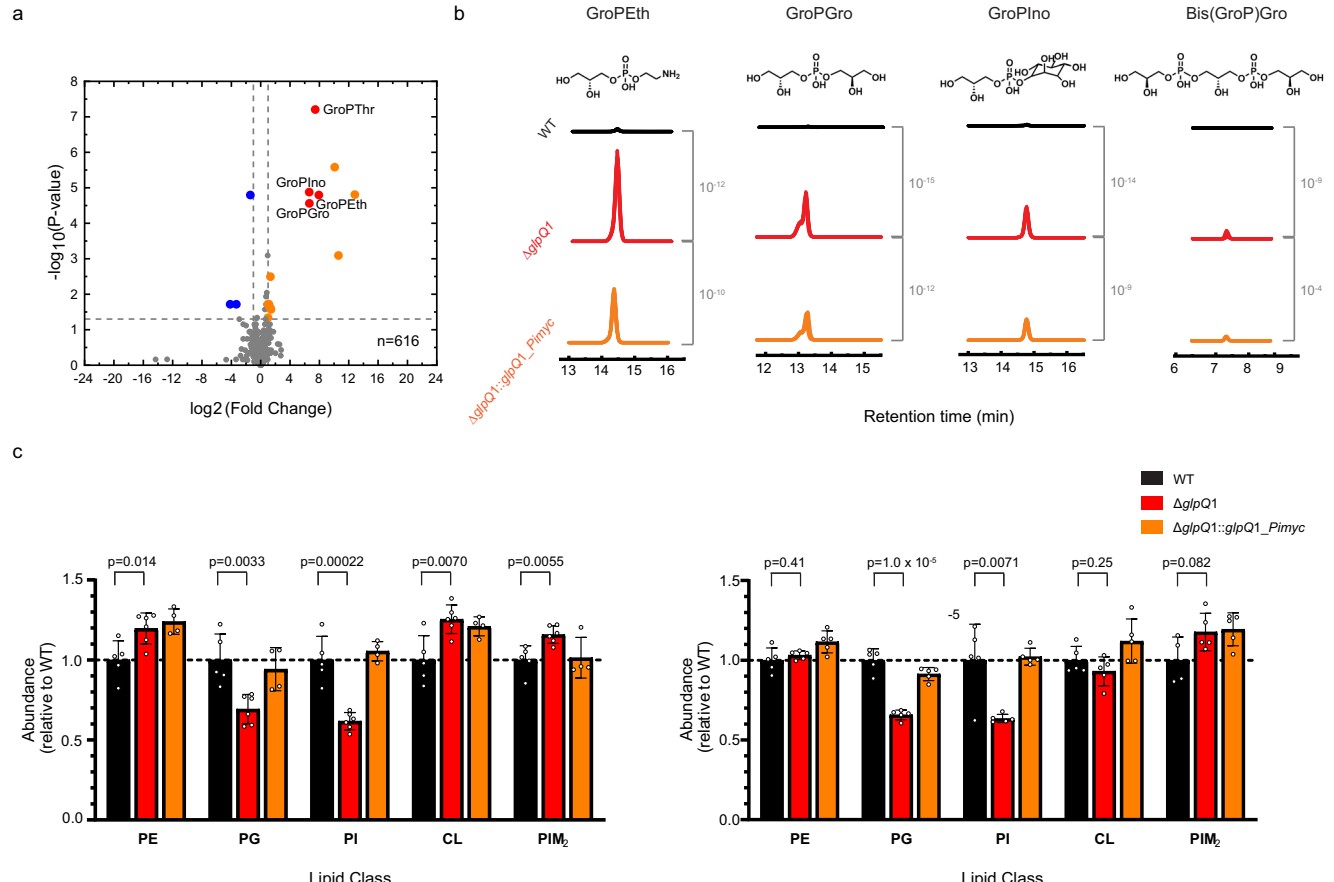

**Fig. 1 | Response of the Mtb metabolome and lipidome to *glpQ*1 deletion.**
**a** Volcano plot showing the metabolome of Mtb plotted as the fold change in the mean abundance of each feature in Δ*glpQ*1/WT, showing negative ion-mode. Features in orange and red are statistically significantly enriched in the Δ*glpQ*1 mutant and represent unidentified and identified features respectively, features in blue are statistically significantly depleted in the Δ*glpQ*1 mutant. Means were calculated across replicate cultures: Δ*glpQ*1 = 6 replicates, WT = 8 replicates. Representative of two independent experiments. *P* values are from two-sided Student's *t*-test, and are adjusted for multiple comparisons setting a false discovery rate of 5%. **b** Extracted ion chromatograms for each of the four lipid-heads of Mtb, shown beneath the chemical structure of the neutral molecule. Note that the chromatogram for each replicate culture within each strain are overlain. *P* values shown in grey refer to the level of significance of the changes in mean intensity for the ion between the strains indicated. **c** Plots of the abundance of the five major phospholipid classes of Mtb in

each strain, as found by lipidomic analysis. The bars represent the mean abundance of the lipid class in each strain, drawn as relative to the level in the WT (mean abundance in the WT set to 1). For each lipid class, the 5 most abundant species of lipid were summed. Bar height represents the mean summed abundance across replicate cultures for that strain, the error bars show the standard deviation. Individual points represent the summed abundance for each individual replicate culture of that strain. *P* values are the 2 tail *t*-test comparing the abundance in the Δ*glpQ*1 mutant versus the WT. The plots display two independent experiments side by side (Experiment 1: WT = 5 replicates, Δ*glpQ*1 = 6 replicates, Δ*glpQ*1::*glpQ*1_*Pimyc* = 4 replicates. Experiment 2: WT = 5 replicates, except for PIM₂ = 4 replicates, Δ*glpQ*1 = 5 replicates, Δ*glpQ*1::*glpQ*1_*Pimyc* = 5 replicates). Data shown is from positive ion-mode MS, except for PIM₂ which is negative ion-mode. Source data are provided as a Source Data file.

these phospholipids. Therefore, the accumulation of GroPCho in Δ*glpQ*1 is likely the result of Mtb taking up GroPCho derived from PC in the media, which is then hydrolysed by GlpQ1 in the WT (Fig. 2a).

We hypothesized that GlpQ1 degradation of host-derived lipid-heads might provide an alternative source of phosphate to allow Mtb to grow and divide. To test this hypothesis, we transferred exponentially growing bacteria into a modified Pi-free Middlebrook 7H9 medium. Standard Middlebrook medium contains 25 mM Pi, which is supraphysiological to the <10 μM Pi present in macrophage phagosomes. After a 72-h period of preconditioning, WT, Δ*glpQ*1 and Δ*glpQ*1::*glpQ*1_*Pimyc* strains were grown in Pi-free media. Surprisingly, all three strains could grow in Pi-free media for around 8 days, with no substantial difference in growth between the strains (Fig. 2d and Supplementary Fig. 4). This indicates that Mtb must have mobilizable intracellular phosphate stores. In Pi-free culture, the growth plateau was reached at a much lower optical density than in the phosphate-replete (25 mM) medium control arm. Next, we transferred bacteria which had reached growth stasis in Pi-free media into fresh Pi-free media supplemented with 25 mM

GroPCho. As negative and positive controls, bacteria were also transferred into fresh Pi-free media and into Pi-replete (25 mM) media, respectively. No growth was observed in fresh Pi-free media, whereas exponential growth restarted in 25 mM Pi culture after a lag of 3–4 days, demonstrating that growth stasis was Pi depletion. The bacteria transferred into 25 mM GroPCho, started to divide after a longer lag period of 6 days and at a much slower rate than the bacteria transferred into 25 mM Pi media, reaching a lower plateau at around 20–25 days post transfer. WT Mtb grew better on this substrate than Δ*glpQ*1, with Δ*glpQ*1::*glpQ*1_*Pimyc* displaying intermediate growth, consistent across two independent experiments. These findings demonstrate that Mtb can access exogenous GroPCho as a sole source of phosphate to reactivate cell division and that this process is partly GlpQ1 dependent.

As GroPCho is the lipid-head of dipalmitoylphosphatidylcholine (DPPC), which is PC (16:0/16:0), the main lipid component of human pulmonary surfactant, and that the major site of surfactant breakdown in normal pulmonary physiology is within the phagosome of the alveolar macrophage[31–33], we propose that GlpQ1 mediated production

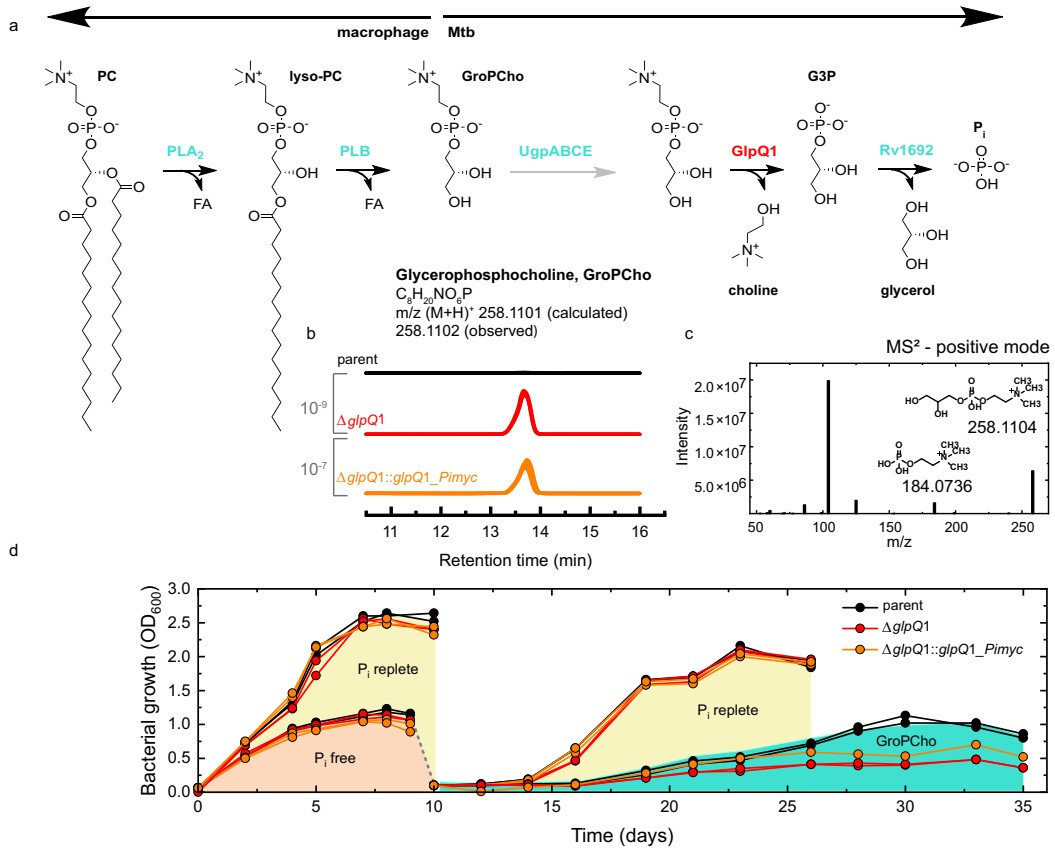

**Fig. 2 | Discovery of glycerophosphocholine as a host-derived alternative phosphate source. a** The pathway from host phosphatidylcholine (PC, shown on the left of the reaction), to phosphate (Pi shown on the right), with UgpABCE actively importing the glycerophosphocholine intermediate into the mycobacterium. PLA$_2$: phospholipase A2. PLB: phospholipase B. **b** Overlay of the extracted ion chromatograms for glycerophosphocholine (M+H)$^+$ of each replicate culture for each strain, black = WT (6 replicates), red = ΔglpQ1 (5 replicates shown), orange = ΔglpQ1::glpQ1-Pimyc (5 replicates shown). In the overlays, 1 outlying replicate was removed from the ΔglpQ1 and ΔglpQ1::glpQ1_Pimyc strains. P values (2 tail t-test) are shown in grey, calculated from all 6 replicate cultures per strain. **c** HCD spectrum at the level of MS$^2$ for glycerophosphocholine, (M+H)$^+$, in positive ion-mode. The peak at 184 is diagnostic. **d** Glycerophosphocholine sole phosphate source experiment. Growth profiles of the strains as labelled in Pi-replete (25 mM) and Pi-free (0 mM) media. At day 9 bacteria were transferred from Pi-free media into fresh Pi-replete media or into GroPCho media (0 mM Pi, 25 mM glycerophosphocholine). Duplicate cultures were performed per strain. Source data are provided as a Source Data file.

of phosphate from DPPC derived exogenous GroPCho may be an important nutritional pathway for Mtb residing within the phagosome during pulmonary infection.

No consensus on the importance of glpQ1 (rv3842c) during infection can be drawn from transposon mutagenesis studies; that is, while DeJesus et al.[34] concluded that disruption of rv3842c led to a growth advantage in vitro, Sassetti et al.[35] indicated that mutants could survive during murine infection, in contrast, Rengarajan et al.[2] demonstrated that the gene is required for survival in murine macrophages. The titratable CRISPRi study of Bosch et al.[36] scored glpQ1 with low vulnerability to disruption. We therefore investigated the performance of our clean genetic deletion, ΔglpQ1, in the low-dose aerosol infection mouse model of TB (Supplementary Fig. 5a–c). A 2 log$_{10}$ reduction in pulmonary colony-forming units versus the parent strain was observed in the initial experiment. In the second and third independent experiments, the phenotype in ΔglpQ1 was similar, but of lower magnitude. To ensure that these changes observed were not confounded due to loss of PDIMs (phthiocerol dimycocerosates) from any strain, a process known to impact upon Mtb virulence studies i.e.[37], PDIM extraction and thin layer chromatography (TLC) analysis was performed (Supplementary Fig. 5d), which showed generally similar PDIM levels across the strains tested.

However, although attenuation was seen consistently with the ΔglpQ1 strain versus the parent strain across all three experiments, complementation was absent with both of the ΔglpQ1::glpQ1_Pimyc and ΔglpQ1::glpQ1_Psmyc strains. This experiment suggests that GlpQ1

is important for Mtb infection, but the lack of complementation means the point remains inconclusive.

## Mtb extensively remodels its envelope when grown without phosphate

Producing an envelope for daughter cells when dividing in Pi-free conditions poses a substantial phosphate requirement. To overcome this, we hypothesised that Mtb might remodel its lipidome, replacing phospholipids with alternative phosphorus-free lipid classes. To address this, WT, ΔglpQ1 and ΔglpQ1::glpQ1_Pimyc were cultured in Pi-free medium until growth plateau, then their LCMS lipidome profiles were compared to those of the same strains grown in a 25 mM Pi-replete media. Remarkably, WT Mtb profoundly remodeled its lipidome when cultured in Pi-free medium (Fig. 3a, b and Supplementary Fig. 6a, b). Of 2963 lipids detected in positive ion-mode, 52% were significantly altered in the WT when grown in Pi-free vs. in Pi-replete (25 mM) media. Remodeling was roughly symmetrical, with 878 features depleted in Pi-free culture and 674 enriched. Fold-changes for the most altered lipids exceeded 2$^{20}$ and the changes were sufficiently extensive as to be visible at the level of the total ion chromatogram (Supplementary Fig. 6a), with a particularly prominent increase in lipid abundance in the retention time range of 21 and 24 min in Pi-free medium-derived lipidome.

Furthermore, annotation of significantly altered features based upon accurate mass, retention time and HCD spectra confirmed the

phenomenon of phospholipids being depleted and replaced with phosphorus-free lipid species. These changes were also seen in the $\Delta glpQ1$ and $\Delta glpQ1::glpQ1\_Pimyc$ strains, indicating that the process is GlpQ independent (Fig. 3b). This independence on GlpQ is expected, as there are other routes for phospholipid catabolism (e.g., polar head hydrolysis via phospholipases C and D) and the biosynthesis of phosphorous-free lipids should be independent of GlpQ.

### Substitution of phospholipids by phosphorous-free lipids

In Pi-free culture, the WT strain demonstrated a 73% overall reduction in the abundance of the 61 detected phospholipid species (Supplementary Fig. 6c), and this figure was remarkably consistent between independent experiments. PIM$_2$s, which also contain a phosphate group, were reduced by 33% (Supplementary Fig. 6b–d). Again, this illustrates a relative sparing of PIM$_2$ versus PI levels, as PI as a class reduced by 78%, and suggests shunting of remaining PI to preserve PIM$_2$s in the PM. Finally, mannosyl phosphomycoketides (MPMs), a class of mycobacterial phosphate-containing lipids known to be immunogenic and recognized by CD1c-restricted T-cells[38,39] also proved phosphate sensitive, decreasing by 18-fold in Pi-free culture (Fig. 3b, Supplementary Fig. 6b,c,e,f). Overall, for the 70 phosphate-containing lipid species detected, there was a decrease of abundance of 70% when Mtb was grown in Pi-free medium, demonstrating that Mtb can substantially reduce the phosphate content of its cell

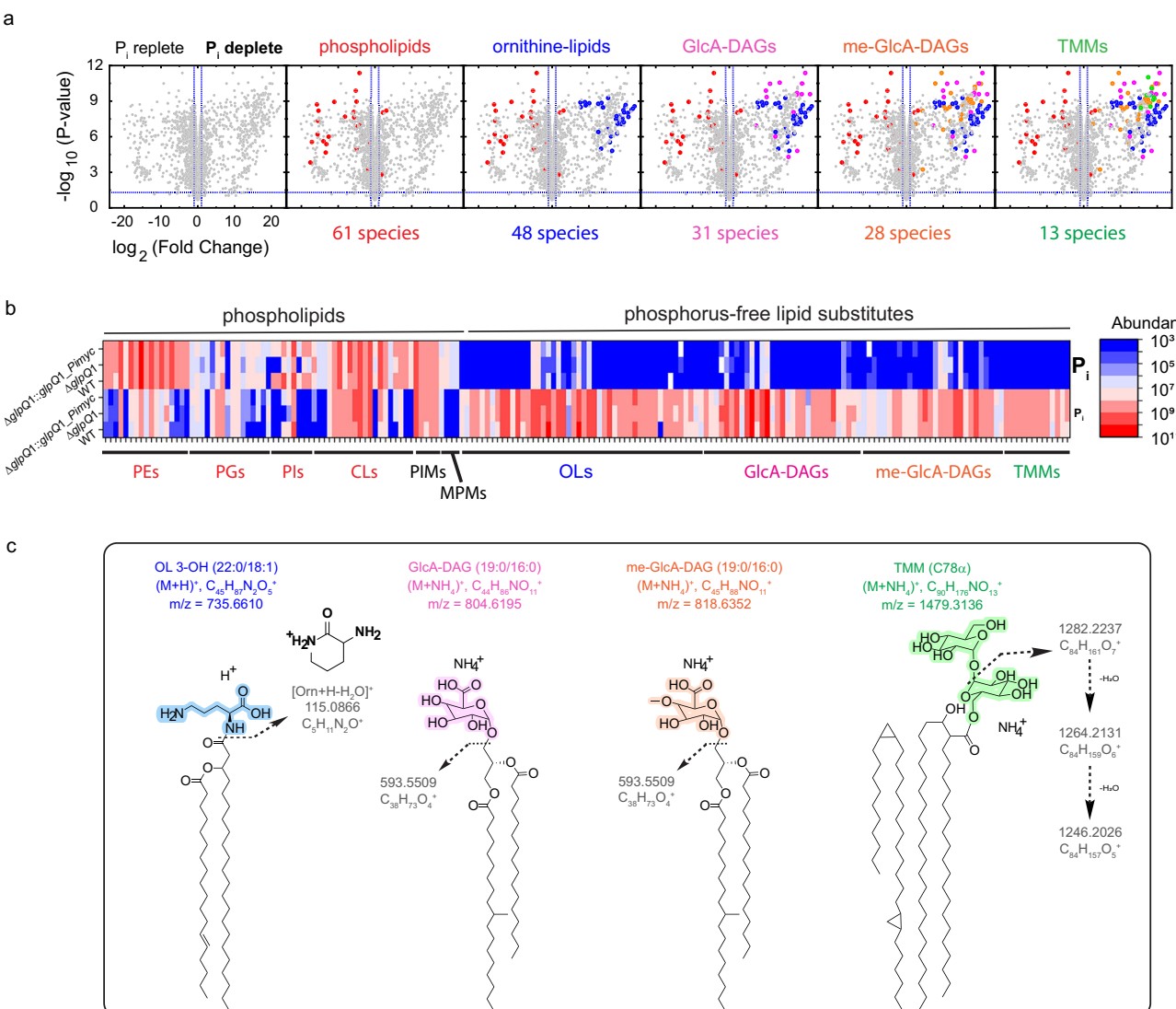

**Fig. 3 | Remodelling of the Mtb lipidome in response to starvation. a** Volcano plot showing the lipidome of Mtb, plotted as the fold change in the mean abundance of each detected lipid in the WT grown in Pi-free media/ mean abundance in the WT grown in Pi-replete (25 mM) media, positive ion-mode data. Thus, features to the left of the midline are lipids enriched in replete Pi and features to the right are lipids enriched in zero Pi. Left to right volcanoes stepwise annotate the major classes of altered lipids. Total features plotted = 2963. Means are calculated across 6 replicate cultures (Pi-free culture) and 5 replicates cultures (Pi-replete culture). A CV < 30 in the PBQCs statistical filter has been applied. *P* values are from two-sided Student's *t*-test, and are adjusted for multiple comparisons setting a false discovery rate to 5%. Representative of 2 independent experiments. **b** Heatmap of the abundance of the 181 lipid species comprising the phospholipids and phosphorus-free lipids detected in positive ion-mode, as well as 4 MPMs and 5 PIMs detected in negative ion-mode. Top 3 rows depict the mean abundance of each lipid (area under the curve) in each strain grown in Pi-replete media (25 mM), bottom 3 rows each strain grown in Pi-free media. Lipid species are organised on the *x*-axis into their classes. Means are calculated across replicate cultures per strain. Representative of two independent experiments. **c** Chemical structures for an example of each of the four classes of phosphorus-free replacement lipids. In each example, the numbers in backets (W:X/Y:Z) indicate W carbon length and X number of unsaturations in the one alkyl chain, and Y and Z the same features respectively in the second alkyl chain. For TMM C78α refers to the 78 carbons in the mycolate chain and its alpha configuration. The polar headgroups are coloured, and arrows indicate the major fragmentation lines demonstrated in the HCD data enabling their annotation. Source data are provided as a Source Data file.

envelope when challenged by Pi scarcity, sustaining cellular physiology and cell-division for longer.

## Identification of the phosphorous-free Mtb lipidome

To maintain cell envelope area and thus cell size and viability, depleted phospholipids must be replaced with alternative, phosphorus-free lipids with similar properties. By using MS[2] spectral data, we were able to identify four major classes of phosphorus-free lipids which were markedly enriched in Mtb in response to phosphate starvation, thus acting as substitution lipids (Fig. 3c and Supplementary Fig. 6g).

Ornithine lipids (OLs) have previously been described to replace phospholipids in response to Pi limitation in bacteria, in particular in environmental species[7,10,40–44]. OLs containing a β-hydroxy fatty acid in an amide linkage with the α-amino group of ornithine, and the second acyl chain esterified at the β-hydroxy position (Fig. 3c), are one of the most common OLs. OLs have been described before in Mtb[45,46], but when Mtb is cultured in phosphate-replete conditions their abundance is negligible. Furthermore, these lipids have seldom been studied in mycobacteria[24] and do not appear in the mycobacterial lipid reference database[29]. In this study, using the presence of a diagnostic HCD fragment ion of $m/z$ 115.0866 which corresponds to a 3-amino-2-oxopiperidium ion (Fig. 3c and ref. [47]), we confidently annotated 48 species of OL in Mtb grown in Pi-free medium. By contrast, only 10 of these species were detectable in Pi-replete cultured bacteria, and the summed total area under the curve for the OL class showed a dramatic 1020-fold enrichment in the lipidome obtained from Mtb grown in Pi-free medium versus Pi-replete culture Mtb.

Next, we annotated glucuronic acid-containing lipids, glucuronyl-diacylglycerols (GlcA-DAGs). We detected 31 species of GlcA-DAGs as ammoniated ions in our data from the Mtb cultured in Pi-free medium, with only 6 detectable in the Mtb culture in Pi-replete medium. The summed total abundance for GlcA-DAGs showed a mean 4100-fold increase in lipidomes from Pi-free vs. Pi-replete cultures. HCD MS[2] confirmed the identity of this lipids class, with both a neutral loss of 211 in positive ion-mode originated from the ammoniated ions (Fig. 3c), which represents the loss of the glucuronic acid head group (193) and the ammonium (18)[48], and also a characteristic fragment ion in the MS[2] spectrum of the deprotonated ion, of mass $m/z$ 249.06, corresponding to a glucuronylglycerol moiety containing an oxirane group[17,49]. This membrane glycolipid has previously been described in *M. smegmatis*[50], but never studied in Mtb. Again, our data show that this usually minor, phosphorus-free lipid becomes a dominant lipid class when Mtb is challenged by Pi restriction.

We subsequently found a cluster of 28 lipid features that were markedly enriched in lipidomes from Pi-free cultured bacteria and exhibited a retention time of around 4.5 min (Fig. 3a–c). Examination of their MS[2] spectra in positive-ion mode indicated that they produced very similar fragmentation to GlcA-DAGs, except the neutral loss seen was now $m/z$ 225 (Fig. 3c). We hypothesised that as in the case of GlcA-DAGs, this represented the loss of the ammonium adduct and the head group sugar, but rather than glucuronic acid ($m/z$ 193), this moiety had an $m/z$ 207. This difference of +14 is consistent with a single O-methylation of the glucuronic acid headgroup in these lipids. Therefore, these results represent a novel class of glycolipids, O-methyl glucuronyl-diacylglycerols, or me-GlcA-DAGs.

Methylation of glycans in nature is considered to be rare[51], and completely absent in mammals, including humans, and therefore a conserved target for immune recognition and defence[52]. However, two arguments support this annotation. First, the observed retention time shift in the diol column from 21 min for GlcA-DAGs to 4.5 min for me-GlcA-DAGs is consistent with the masking of a polar hydroxyl group with a non-polar methyl, which in the diol column should elute earlier. Second, O-methylated glucuronic acid has been previously documented in several mycobacteria, for instance as a constituent of a pentasaccharide hapten in *Mycobacterium avium*[53] as well as in

glycopeptidolipids in *Mycobacterium habana*[54]. We therefore consider this annotation as highly plausible.

Assuming our annotations are correct, of 28 putative me-GlcA-DAGs in the Mtb grown in Pi-free medium only 10 were also detectable in the WT grown in Pi-replete media. MS[2] spectra were available for 22 species, all of which showed the neutral loss of 225. The summed total abundance for all 28 me-GlcA-DAGs combined showed a 120-fold increase in Mtb cultured in Pi-free medium vs. in Pi-replete medium. Therefore, me-GlcA-DAGs may be a third phosphorus-free PM lipid which Mtb uses to substitute in for phospholipids.

Surprisingly, we also observed a significant change in trehalose monomycolates (TMMs). These lipids contain long chain mycolic acids, are phosphorus-free, are an essential component of the mycobacterial envelope[55], and have established fragmentation patterns[29], enabling confident annotation. We detected 13 distinct TMMs as ammoniated ions in Mtb grown in Pi-free medium with each species exhibiting ion intensities of between $10^7$ and $10^8$ counts, whereas none of these lipids were detected in Mtb grown in Pi replete culture (Fig. 3a–c and Supplementary Fig. 6g).

Therefore, during challenge by phosphate restriction, the cell envelope of Mtb undertakes a marked remodeling, with phosphorus-free lipid species which are barely detectable when Pi is freely available becoming dominant classes, married to a reduction in both simple and decorated phospholipids. As Mtb remains viable in phosphate-free media, the altered lipidome must still support a functioning cell envelope. To investigate how the cell envelope might physically alter, we performed cryogenic electron microscopy analysis of WT Mtb grown in Pi-free and Pi-replete media (Fig. 4a–c). We found that the cell envelope thickness in the Pi-free condition was both significantly thicker ($p < 0.0001$), and showed a greater variance, compared to in Pi-replete cultured bacteria (Fig. 4c). These findings provide orthogonal evidence for functional lipidome remodeling in response to Pi restriction, reflecting the changes in lipid composition characterized by LC-MS.

As such conditions are thought to be encountered within the macrophage phagosome during infection, this Pi-restricted lipidome can be considered as a more physiological makeup. These changes are represented in Fig. 4f.

## A synthetic standard supports the presence of phosphorus-free lipids

None of the four phosphorus-free lipid species which we found to be upregulated under Pi depletion is commercially available to use as an LCMS standard, an important experimental validation for novel species. Therefore, we synthesized one of the most abundant species in our Mtb extracts, GlcA-DAG (16:0/16:0)[49] (Fig. 5a and Supplementary Fig. 7). We analyzed the synthetic GlcA-DAG (16:0/16:0) lipid standard, which demonstrated comparable negative and positive ion-mode MS[2] spectral profiles as the same feature in our lipid extracts of Mtb (Fig. 5b, c). To conclusively prove our annotation of this class of lipid in our Mtb extracts, we analyzed an extract from Mtb grown in Pi-free medium, before and after spiking it with synthetic GlcA-DAG (16:0/16:0) to a final concentration of 50 μg/mL. As shown in Fig. 5d, the extracted ion chromatogram (EIC) for GlcA-DAG 16:0/16:0 (of $m/z$ 762.5726) augmented appropriately in the spiked extract, demonstrating co-elution and therefore confirming the annotation.

We next leveraged the GlcA-DAG (16:0/16:0) synthetic standard to provide further evidence for the me-GlcA-DAG class in Mtb. It is recognised that extraction from biological samples using methanol-containing solvent mixtures may lead to spurious methylation[56]. We therefore stored the synthetic GlcA-DAG (16:0/16:0) standard in the solvent mixture used in our lipid extraction protocol, 2:1 (v/v) chloroform:methanol, for 48 h, and then analyzed it with LCMS. We could not detect a peak corresponding to a me-GlcA DAG (16:0/16:0) in this sample. Furthermore, spiking of our Mtb cell extract with synthetic

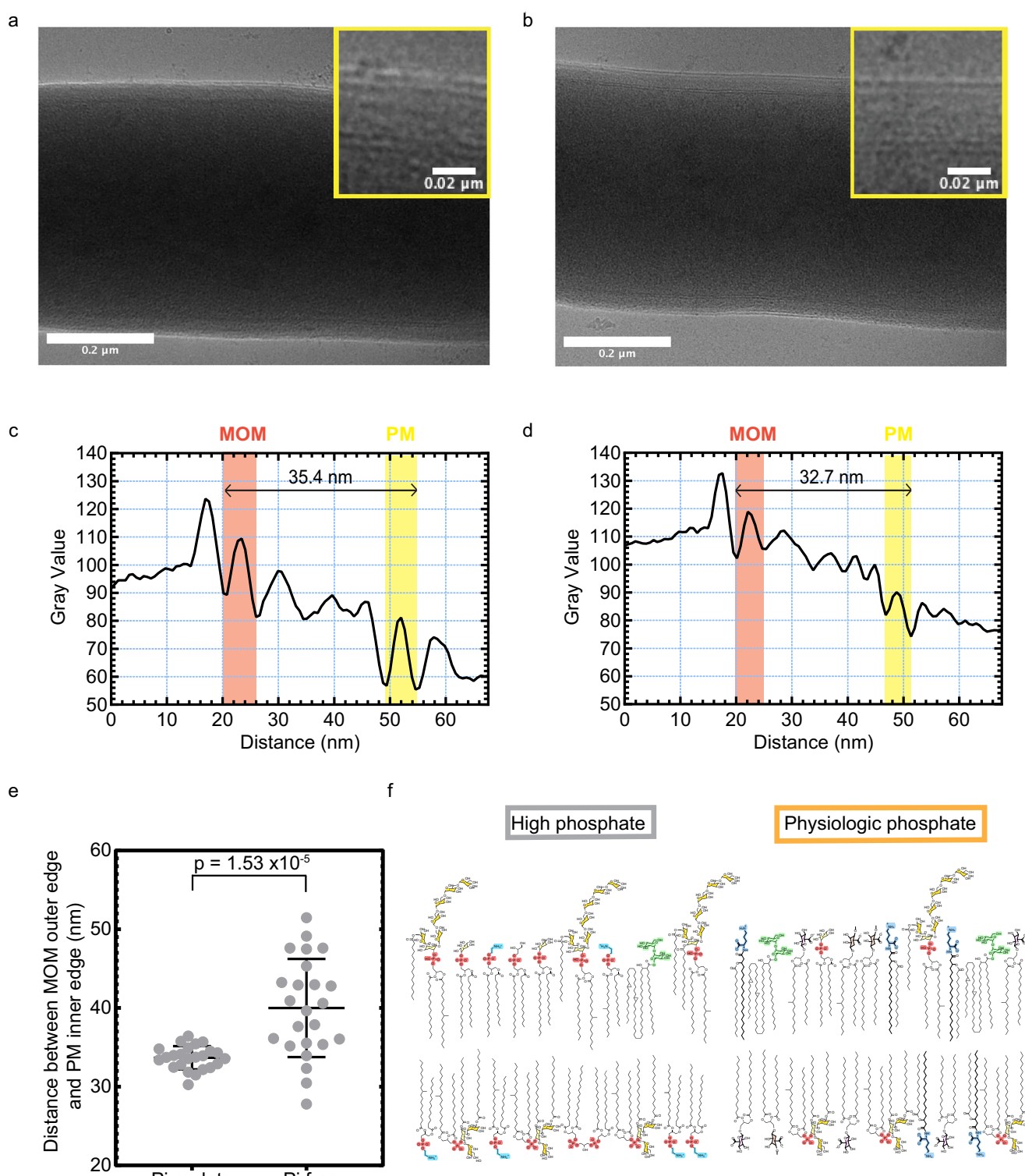

GlcA-DAG (16:0/16:0) in 2:1 (v/v) chloroform:methanol to 50 μg/mL did not lead to augmentation of the EIC for the ammoniated ion of me-GlcA DAG (16:0/16:0) (Fig. 5d). It is therefore unlikely that the proposed me-GlcA-DAG species detected in our Mtb lipidomic experiments are artefactual due to methanol extraction. Further support of this conclusion is provided by the finding that for each diacylglycerol alkylform, the ratio of abundance of the proposed me-GlcA-DAG to the corresponding GlcA-DAG in our Mtb extracts is non constant (Fig. 5e). Thus, an in vivo enzymatic methylation by a currently unknown methyltransferase must generate these lipids. Supplementary Data 3

details the ion features for all 190 lipid species analyzed in 0 mM and 25 mM Pi cultures, and Supplementary Notes 1 the HCD spectra for an example from each lipid class.

## Discussion

Growing bacteria outside of their natural environment poses a fundamental challenge, a metabolic disconnect with reality. In the case of Mtb, the use of media such as Middlebrook 7H9 broth and 7H10/11 agar are rich in glycerol, which is an important carbon source for the bacteria in vitro, but absent in vivo[57], and Pi is present in

**Fig. 4 | Envelope remodelling to phosphate restriction is reflected by changes in envelope thickness. a** Cryo-EM image of a WT Mtb bacillus cultured in Pi free media. Inset shows a greater magnification of the cell envelope at the cell margin at the top of the image. **b** Equivalent Cryo-EM image of WT Mtb bacillus cultured in 25 mM Pi media. **c** Density plot showing the measurement of the cell envelope thickness of the bacillus in a. The measurement was made from the outermost extent of the mycobacterial outer membrane (MOM), to the innermost extent of the plasma membrane (PM) as indicated by the arrow. See *methods* for details. Each bacillus studied had envelope measurements taken at 5 different positions. **d** Equivalent density plot for the cell envelope of the bacillus shown in (**b**). **e** Plot of the envelope measurements for WT bacilli grown in Pi free and Pi replete (25 mM)

media. Points represent the mean of five measurements per an individual bacillus studied, horizontal bars the means of the 24 bacilli measured per condition, and error bars the standard deviation. *P* value is from 2-sided Student's *t*-test comparing the mean measurements of the envelope thickness taken of each bacillus, from the two populations. **f** Diagram of the PM of wild-type Mtb when phosphate is present in excess (left) and when phosphate is restricted (right). Phosphorus-free lipids are drawn with their polar headgroups colour coded as per Fig. 3c. Phospholipids are represented as per Supplementary Fig. 1b. Mono- and di- acylated $PIM_2$ and $PIM_6$ are drawn with the mannose sugars coloured in gold and inositol in light yellow, and are labelled in Supplementary Fig. 1c. Source data are provided as a Source Data file.

supraphysiologic amounts, serving as the pH buffer. Our findings reveal that Pi availability has a profound effect on the composition of mycobacterial lipidomes and membranes.

We characterized extensive lipid remodeling in response to Pi starvation, which revealed two mechanisms by which Mtb overcomes this limitation to cell division. First, Mtb can access host-produced phospholipid polar heads, in particular GroPCho, as a source of phosphate, in a GlpQ1-dependent manner. Second, Mtb can extensively replace its PM phospholipids with phosphorus-free lipid species. Both mechanisms are likely important in macrophage infection (where Pi concentrations are below 10 μM) during the pathogenesis of pulmonary TB.

Based on our findings, we propose a model whereby Mtb resides in the phagosomes of alveolar macrophages during human pulmonary infection, which have limited Pi but are rich in GroPCho. GroPCho is produced from DPPC, the principal phospholipid in pulmonary surfactant, which is itself taken up into alveolar macrophages by phagocytosis as part of normal pulmonary surfactant metabolism[31–33]. Either host or secreted Mtb phospholipases may hydrolyse DPPC to GroPCho. After active uptake into phagosome-residing Mtb by the ABC transporter UgpABCE, GlpQ1 hydrolyses host-derived GroPCho to provide an alternative source of phosphate. The Mtb genome encodes only 4 ATP-powered transporters specific for carbohydrates in its PM, in marked contrast to *M. smegmatis*, which contains 19.[58,59] This is postulated to be a consequence of Mtb encountering the carbohydrate-poor environment of the phagosome during infection. That Mtb contains the UgpABCE transporter specific for glycerophosphodiesters therefore suggests this is an important source of nutrition during such macrophage infection.

There is evidence that *Staphylococcus aureus* uses an analogous mechanism to acquire phosphate during host infection[60]. Employing a secreted glycerophosphodiesterase, the pathogen was proposed to derive glycerol-3 phosphate from teichoic acids in the cell walls of bystander bacteria, to utilize as a phosphate source whilst inside the Pi-poor environment of the human nose. However, ours is the first study to interrogate GlpQ1 in a mycobacterial species, and the potential co-localization during pulmonary infection of Mtb, Pi restriction and DPPC abundance within the alveolar macrophage phagosome is intriguing. This, coupled with the demonstrated upregulation of the expression of all four of the genes of the glycerophosphodiester transporter, UgpABCE, in Mtb in response to the overexpression of *regX*3 (the Mtb low phosphate response regulator)[21] makes such a model compelling. Interestingly, nuclear magnetic resonance studies have shown PC levels decrease and GroPCho levels increase over time inside lung granulomas in Mtb infected guinea pigs[61], spatially linking this biochemical process to pulmonary infection. It would be desirable to study the GlpQ1 enzyme and the UgpABCE transporter in tandem, as the nutritional pathway could be regulated at either level, and indeed upregulation of the transporter could partially compensate for knock-out of the enzyme in the glpQ1 mutant.

During Pi restriction, Mtb can generate ATP from cytosolic polyphosphate stores and this mechanism may be required for the full virulence of Mtb in guinea pigs and for survival in THP-1

macrophages[62]. In parallel, we show here that Mtb reduces the amounts of PE, PI, CL, PIMs and MPMs in its membranes. In this way, the cell envelope may be considered a second accessible phosphate storage compartment for Mtb, resulting in phospholipids being replaced by phosphorus-free lipids. These lipids are upregulated to maintain a viable PM, but the observed increase in TMMs, which shuttle mycolic acids into the outer mycomembrane[63], suggests that remodeling in response to low Pi extends beyond the PM.

Our findings are important for understanding the composition of the Mtb cell envelope under physiologic conditions that reflect Pi concentrations in alveolar macrophages. We show that >50% of lipids in the Mtb lipidome are significantly altered during culture in Pi-free media, with $2^{20}$ fold increases in individual lipids. This previously unrecognized aspect of the Pi starvation response involving phospholipid substitution could allow Mtb survival during the early phases of infection and may also modulate immunogenicity of the pathogen. Thus, culturing Mtb in Pi-replete media, which is the current standard, hinder experiments aimed at developing vaccines and antibiotics against TB. For example, there are efforts currently to produce lipid-based vaccines against Mtb[64–68], targeting mycobacterial lipids that can stimulate human CD1-restricted T cells. As lipids known to date to be presented by CD1 molecules to T cells may not necessarily be of high abundance in the Pi-starved lipidome, our findings should be considered in this field. MPMs are a direct example of this.

The contribution of CD1-restricted T cells reactive to lipids that are enriched in the phosphate-starved lipidome identified here, to the human immune response to Mtb infection, warrants further investigation, especially as *M. smegmatis* GlcA-DAGs have already been demonstrated to stimulate CD1d-restricted natural killer T cells[69] and Mtb TMMs have been recently shown to stimulate CD1b-restricted human T cells[70]. Additionally, identifying the biosynthetic enzymes responsible for assembling phosphorous-free lipids during phosphate restriction could reveal targets for future drug discovery.

## Methods

### Strains

To generate the *glpQ1* knock-out strain from the parent H37Rv, an unmarked, in-frame deletion of the *rv3842c* gene was made using the method of Parish and Stoker[71]. Briefly, 1.5 kb of flanking sequence of the genes *rv3842c* (*glpQ1*) was amplified from the genomic DNA of H37Rv with KAPA HiFI Hotstart taq using the primers *Rv3842c* 5′ for and *Rv3842c* 5′ rev for the 5′ side and *Rv3842c* 3′ for and Rv3842c 3′rev for the 3′ side (see Table 1 below). The products were cloned into pCR4blunt (Invitrogen) and sequenced. The inserts were digested with HindIII and XbaI and the fragments ligated together using T4 DNA ligase. The ligation product was amplified using primers Rv3842c 5′ for and Rv3842c 3′ rev and the fragment was cut with HindIII and cloned into the HindIII site of p2NIL. The PacI fragment of pGOAL 17 containing the *lacZ* and *sacB* genes was cloned into the PacI site of the resulting plasmid to make the ΔglpQ1 construct. Competent H37Rv cells were prepared from cells grown to an $OD_{600}$ of 1 and washed with 10% glycerol and electroporated with 2 μL plasmid. Single crossovers were selected on 7H11 plates containing kanamycin and Xgal. The blue

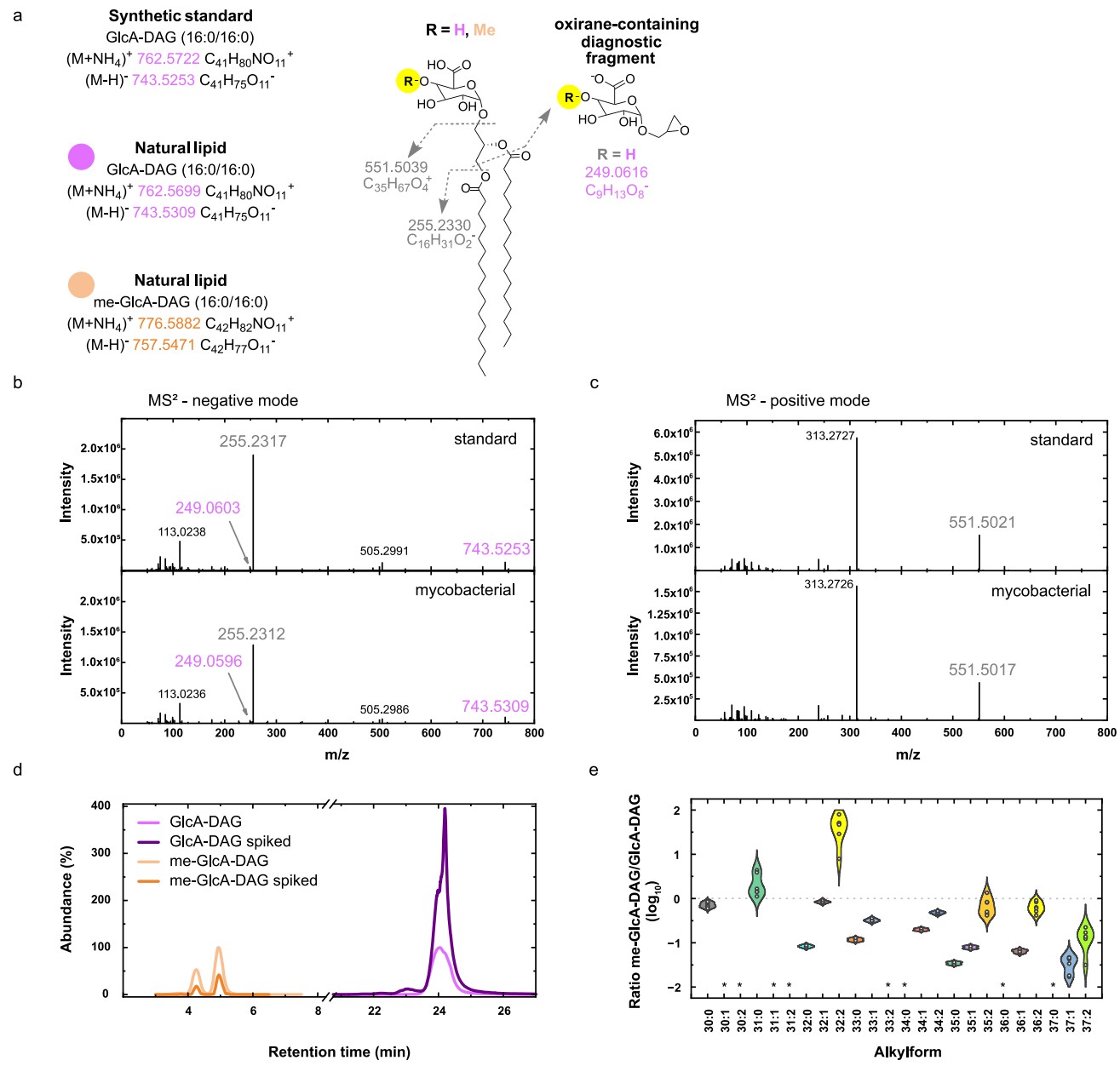

**Fig. 5 | Use of synthetic standard to confirm annotation of glucuronyl-diacylglycerols in Mtb. a** Structure of the 16:0/16:0 alkylforms of GlcA-DAG and me-GlcA-DAG, demonstrating the HCD product ions shared by the two lipids (grey) as well as the oxirane-containing diagnostic fragment produced in negative ion-mode MS² for GlcA-DAGs (me-GlcA-DAGs are not detected in negative ion-mode). The methylation is shown at the 4 OH position of the glucuronic acid sugar, denoted as R, but the position is not confirmed. **b** Negative ion-mode HCD MS² spectra for the deprotonated ion of GlcA-DAG (16:0/16:0) comparing that for the synthetic lipid (top) to that obtained from the mycobacterial lipid extract (bottom). The diagnostic oxirane-containing fragment of *m/z* 249 is indicated. **c** Positive ion-mode MS2 spectra for the ammoniated ion of the same. Note that due to its complete fragmentation at this collision energy, the precursor ion of

762.5726 is not visible. **d** Positive ion-mode EICs for the ammoniated ions of me-GlcA-DAG (16:0/16:0) and GlcA-DAG (16:0/16:0) from WT Mtb grown in Pi-free media, before and after spiking with synthetic GlcA-DAG (16:0/16:0) to 50 μg/mL. The synthetic lipid had been stored in 2:1 (v/v) chloroform: methanol for 48 h prior to spiking. The bifid shape of the peak for the me-GlcA-DAG may indicate that isotypes of this lipid exist with the methylation in 2 different positions upon the sugar, altering the retention time slightly. **e** Plot showing the ratio of the abundance of the methylated vs non-methylated glucuronyl-diacylglycerol for each diacylglycerol alkylform detected in Mtb lipid extracts from the WT grown in Pi-free media. The ratio is expressed on a $\log_{10}$ scale. Individual points represent the ratio for individual replicate cultures. Asterisks indicate alkylforms where one or both lipids were not detected. Source data are provided as a Source Data file.

colonies were then streaked on 7H11 plates containing sucrose and Xgal, and the resulting white colonies were screened for double crossovers. Successful gene deletion was confirmed by next-generation sequencing using the Francis Crick Institute's Advanced Sequencing Science Technology Platform.

To generate the initial complement strain, Δ*glpQ*1::*glpQ*1_*Pimyc*, the construct for complementation (pML1335-*Pimyc-rv3842c*) was generated using an in-house plasmid which originated from pML1335, a gift from Michael Niederweis (Addgene plasmid # 32377[72]), which had had the promoter of pML1335 replaced with the intermediate

promoter Pimyc (https://pubmed.ncbi.nlm.nih.gov/15687379/). The *GFPM2*+ insert was replaced with the *rv3842c* gene amplified from Mtb H37Rv genomic DNA. Competent Δ*glpQ1* cells were electroporated with pBS-Int (for integrase) and with pML1335-*Pimyc-rv3842c*. Colonies were screened for successful complementation construct insertion by polymerase chain reaction (PCR) with DNA obtained by InstaGene Matrix (BioRad).

To subsequently generate the complement strain, Δ*glpQ1::glpQ1_Psmyc*, the construct for complementation (pML1335-*Psmyc-rv3842c*) was generated by isothermal assembly (NEBuilder HiFi) using the plasmid pML1335 with its original Psmyc promotor. The *GFPM2*+ insert was replaced with the *RV3842c* gene amplified from Mtb H37Rv genomic DNA. Competent Δ*glpQ1* cells were electroporated with pBS-Int (for integrase) and with pML1335-*Psmyc-rv3842c*. Colonies were screened for successful complementation construct insertion by PCR with DNA obtained by InstaGene Matrix (BioRad).

## LC-MS methods

All solvents used were Optima® HPLC grade from Sigma Aldrich. All work involving viable Mtb were conducted inside the containment level 3 suite, up until the point that a validated method of killing Mtb was carried out.

### HILIC method: ZIC-pHILIC column chromatography – orbitrap MS

Cultures were grown in Middlebrook 7H9 complete media until $OD_{600} = 1$ and then 1 mL transferred onto 0.22 μm filter paper and grown for 5 days on Middlebrook 7H10 media with ADC supplement. After 5 days, cells were scraped from filters into 2:1 (v/v) chloroform: methanol and placed on dry ice for 1 h to inactive the Mtb, with frequent vortexing. Samples were then removed from the containment level 3 laboratory. Samples were then incubated at 4 °C for 1 h with regular sonication, centrifuged at $21,100 \times g$, then the supernatant transferred into an Eppendorf tube and dried under nitrogen stream. The pellet was re-extracted with 450 μL methanol: water (2:1, (v/v), containing 3 nmol 13C5, 15N1-valine for LC-MS), and added to the extract. The dried extract was then suspended a final time in 50 μL chloroform, 150 μL methanol, and 150 μL water. Once settled, the upper phase of this biphasic mixture consisted of the aqueous sample containing the polar metabolites.

The LC-MS method used was adapted from ref. 73 and is as per published previously[74]. Samples were injected into a Dionex UltiMate LC system (Thermo Scientific) with a ZIC-pHILIC column (Merck Sequant). A 15-min elution gradient of 20% solvent B was used, and this was followed by a 5-min wash of 95:5 solvent A to solvent B and 5-min re-equilibration; solvent B was acetonitrile and solvent A was 20 mM ammonium carbonate in water. Other parameters were as follows: flow rate 300 μL/min; column temperature 25 °C; injection volume 10 μL; autosampler temperature 4 °C. MS was performed with positive/negative polarity switching using an Q Exactive Orbitrap (Thermo Scientific) with a HESI II (Heated electrospray ionization) probe. MS parameters were as follows: spray voltage 3.5 kV and 3.2 kV for positive and negative modes, respectively; probe temperature 320 °C; sheath and auxiliary gases were 30 and 5 arbitrary units, respectively; full scan range: 70 to 1050 m/z with settings of auto gain control (AGC) target and resolution as Balanced and High ($3 \times 10^6$ and 70,000), respectively.

## Table 1 | Primers used for *rv3842c* Gene Deletion

| Rv3842c 5' for | GCAAGCTTACTCCTGGTGAATGGTGGGG |
|---|---|
| Rv3842c 5' rev | GCTCTAGAGGCCCATGTCATGTCCACTA |
| Rv3842c 3'for | GCTCTAGAACCACTCGCTGATCCCCGCC |
| Rv3842c 3' rev | GCAAGCTTGCTCGCGGCCCGACCCGATC |

Data were recorded using Xcalibur 3.0.63 software (Thermo Scientific). Mass calibration was performed for both ESI polarities before analysis using the standard Thermo Scientific Calmix solution. To enhance calibration stability, lock-mass correction was also applied to each analytical run using ubiquitous low-mass contaminants. Parallel reaction monitoring (PRM) acquisition parameters: resolution 17,500, auto gain control target $2 \times 10^5$, maximum isolation time 100 ms, isolation window m/z 0.4; collision energies were set individually in HCD (high-energy collisional dissociation) mode. PBQC samples were prepared by pooling equal volumes of each sample and were analyzed throughout the run to provide a measurement of the stability of the system. To confirm the identification of significant features, PBQCs were analyzed in ddMS2 mode. Data were acquired using Xcalibur 3.0.63 (Thermo Fisher Scientific), and the raw files were imported into Progenesis QI v3.0 (Nonlinear Dynamics) which was used for automated retention time alignment and peak detection. Peak picking utilized the software's automatic sensitivity parameter method using the default (level 3) noise estimation algorithm for ion detection. Data from samples were normalized against the total ion abundance. Annotations were initially assigned to accurate masses via the software's automated method with a maximum mass tolerance of 5 ppm using the Chemspider database and confirmed with MS² spectra where available via manual inspection. Freestyle 1.8 SP2 (Thermo Scientific) was used to manually check the integration of all the peaks of the glycerophosphodiesterase species and to further review MS² data. Statistical analyzes were performed in Microsoft Excel after exporting the normalized abundances of each compound from Progenesis as. CSV files.

### Amide method: amide column chromatography – time of flight MS

Cultures were grown in Middlebrook 7H9 complete media until $OD_{600} = 1$, and then 1 ml was transferred onto 0.22 μm filter paper and grown for 5 days on Middlebrook 7H10 media with ADC supplement. After 5 days, cells were scraped from filters into a 1 ml volume of 2:2:1 solution (v/v/v) of acetonitrile: methanol: water with glass beads, and lysed via bead beating with MP FastPrep 24 at 6.5 m/s for 30 s, then filtered using Corning Spin-X centrifuge tubes, then transferred to LC-MS vials.

LC-MS analysis was performed on an Agilent 1290 LC system coupled to an Agilent G6230B ToF mass spectrometer. Chromatography was performed using a Waters XBridge Amide column, 4.6 × 100 mm maintained at 40 °C. Solvent A was 95% 20 mM ammonium hydroxide, 20 mM ammonium acetate, pH 9.0; 5% acetonitrile. Solvent B was acetonitrile. Analytes were eluted using a flow rate of 0.4 mL/min and the following mobile phase gradient: 0 – 3 min, 85 – 30% B; 3 – 12 min, 30 – 2% B; 12 – 15 min, 2% B; 15 – 16 min, 2 – 85% B. The injection volume was 20 μL. The Agilent G6230B ToF was operated in positive and negative polarities with ESI using a dual AJS ESI source. Capillary, nozzle, and fragmentor voltages were set at 3500 V, 2000 V and 110 V, respectively. Data were collected in the m/z range 50–1200 and saved in centroid mode. Dynamic mass axis calibration was achieved by continuous infusion of a reference mass solution, which enabled accurate mass spectral measurements with an error of less than 5 ppm. .d files were imported into Agilent MassHunter Qualitative Analysis B.07.00 where metabolites of interest were identified and quantified using the "Find Compounds by Formula" algorithm with a match tolerance of 10 ppm. Peaks were then manually inspected. Maximum variation in RT for any ion of interest measured between samples was <0.15 min. The obtained peak areas were normalized to the total ion current of each chromatogram which was calculated in Progenesis QI v3.0 after converting .d files to .mznld files using msConvert . Compound measurements were exported from MassHunter as .CSV files and statistical analyzes performed in Microsoft Excel.

## Diol method: BETASIL diol column chromatography-orbitrap MS

For the lipidomic analyzes of exponentially growing bacteria in standard Pi culture conditions, Mtb strains were grown in 4 mL of detergent-free Middlebrook 7H9 complete media until $OD_{600} = 1$. For zero phosphate lipidomics, Mtb strains were preconditioned with 72-h of growth in Pi-free media or 25 mM Pi media, then transferred into fresh media of the same Pi concentration, detergent free, 24 mL volume in 50 mL Falcon™ tubes, and grown until growth stasis-defined as 48-h with static $OD_{600}$ measurements. The composition of the modified Middlebrook media is as details in the *Sole Phosphate Source Experiments* section below.

For extractions, in both experiments, 4 mL of cell culture was then spun for 10 min at $2000 \times g$ and supernatant discarded. For the zero phosphate experiments, at this stage, a proportion of the cell pellet from the cultures grown in 25 mM phosphate was discarded in order to normalize cell mass between the experiment's two arms. The pellets were then washed with Optima® water twice, then extracted in 2:1 (v/v) chloroform: methanol, and incubated on dry ice for 1 h to inactivate the Mtb, with frequent vortexing. Samples were removed from the containment level 3 laboratory and then incubated at 4 °C with regular sonication for 1 h then spun at $21,100 \times g$ for 10 min, and supernatant transferred and dried under nitrogen stream. The pellets were re-extracted with 1500 µL 2:1 (v/v) methanol: chloroform, sonicated, centrifuged at $21,100 \times g$ and added to the dried extracts. Combined lipids were dried under nitrogen and the dried pellet dissolved in 100 µL of 1:1 (v/v) chloroform: methanol.

This cell lipid extract was diluted 1:2 with solvent A (70:30 (v/v), hexanes: isopropanol, 0.02% (m/v) formic acid, 0.01% (m/v) ammonium hydroxide), centrifuged at $300 \times g$ for 5 min to remove trace non-lipidic materials, then transferred to a glass autosampler vial (Agilent). The LC–MS method was adapted from ref. 29. Samples were injected onto a BETASIL diol column (5 µm × 150 mm × 2.1 mm, with BETASIL diol guard column (10 mm × 2.1 mm), held at 20 °C) in an Ultimate 3000 HPLC system coupled to a Thermo Q Exactive Orbitrap MS. Lipids were eluted at 0.15 mL/min with a binary gradient from 0 to 100% solvent B (70:30 (v/v) isopropanol: methanol, 0.02% (m/v) formic acid, 0.01% (m/v) ammonium hydroxide): 0–10 min, 0% B; 17–22 min, 50% B; 30–35 min, 100% B; 40–44 min, 0% B, followed by additional 6 min 0% B post-run. MS data were acquired in both polarities using a full scan method. The positive and negative HESI-II spray voltages were 4.5 and 3.5 kV, respectively; the heated capillary temperature was 250 °C; the sheath gas pressure was 30 psi; the auxiliary gas setting was 20 psi; and the heated vaporizer temperature was 150 °C. The parameters of the full mass scan were as follows: a resolution of 70,000, an auto-gain control target under $3 \times 10^6$, a maximum isolation time of 200 ms, and an $m/z$ range 200–3000. To confirm the identification of significant features, PBQCs were ran in data-dependent top-N (ddMS2-topN) mode; parameters as follows: a resolution of 17,500, an auto gain control target under $2 \times 10^5$, a maximum isolation time of 100 ms, an isolation window of $m/z$ 0.4 and normalized collision energy of 35 V. Data were acquired using Xcalibur 3.0.63 (Thermo Fisher Scientific).raw files were imported into Progenesis QI v3.0 (Nonlinear Dynamics) which was used for automated retention time alignment and peak detection. Peak picking utilized the software's automatic sensitivity parameter method using the default (level 3) noise estimation algorithm for ion detection. Sample data were normalized against the total ion abundance. Preliminary annotations were initially assigned to accurate masses with a maximum error of 12 ppm to the MycoMass database[29], utilizing the Progenesis Meta-Scope identification method, without retention time parameter limits nor fragmentation data. These preliminary identifications were confirmed with retention time and $MS^2$ where available via manual inspection and as described in the text. Freestyle 1.8 SP2 (Thermo Scientific) was used for further analysis, including manual re-

inspection of peaks for features of interest. For TMMs, additional $MS^2$ spectra were performed using the same parameters as above but with collision energy reset in HCD mode to HCD 10.00. After normalizing with total ion abundance, compound measurements were exported from Progenesis QI as .CSV values for statistical analyzes in Microsoft Excel.

### Analysis of synthetic GlcA-DAG (16:0/16:0) standard

The Diol Method was run with total cell lipid extract from WT Mtb grown in Pi-free modified media. One aliquot of this extract was run unspiked, and one aliquot after spiking the extract to a final concentration of 50 µg/ mL with synthetic GlcA-DAG (16:0/16:0) which had been stored first for 48-h in 2:1 (v/v) chloroform: methanol. The two samples were run in the diol-orbitrap LC-MS system, and a gain in the abundance of the ion feature corresponding to the ammoniated ion of GlcA-DAG (16:0/16:0) was observed in positive-ion mode.

Additionally, the total cell extract and the synthetic standard were analyzed side-by-side under the same PRM conditions: resolution 17,500, auto gain control target $2 \times 10^5$, maximum isolation time 100 ms, isolation window $m/z$ 0.4; collision energies were set in HCD mode at HCDs of 10.00, 20.00 and 30.00. Retention times and $MS^2$ spectra were a match in both positive and negative-ion modes.

### Synthesis of GlcA DAG (16:0/16:0)

Synthesis was guided by ref. 49. All solvents used for chromatography were purchased from Fisher Scientific. Flash column chromatography silica cartridges were obtained from Biotage Inc. The purification of mixtures was conducted using Biotage Isolera 1. $^1$H NMR spectra were recorded on a Varian INOVA-500 spectrometer or on a Bruker 400 MHz NMR spectrometer. Chemical shifts (δ) are reported in parts per million (ppm) relative to the residual solvent peak, while coupling constants (J) are reported in hertz (Hz). Waters ACQUITY UPLC H Class Bio coupled with Xevo G2-S QToF with UNIFI was used for high-resolution Mass analysis (HRMS). Refer to Supplementary Fig. 7 for the synthesis scheme.

**(S)−4,4'-(((3-(allyloxy)propane-1,2-diyl)bis(oxy))bis(methylene)) bis(methoxybenzene) (2).** To a solution of (R)−3-(allyloxy)propane-1,2-diol (1, 1.0 g, 7.57 mmol) in DMF (20 mL) was added sodium hydride (470.0 mg, 95%, 19.67 mmol) at 0 °C. After stirring for 15 min, p-methoxybenzyl chloride (3.08 g, 19.67 mmol) and a catalytic amount of tetra-n-butylammonium iodide (TBAI) were added, and the mixture was allowed to warm to room temperature. After stirring overnight at 60 °C, the reaction mixture was quenched with MeOH, and then ethyl acetate was added. The organic solution was washed with water, saturated $NaHCO_3$ and brine, then dried over $MgSO_4$, filtered, and concentrated. The residue was applied onto a column of silica gel. Elution with ethyl acetate/hexane and subsequent concentration of the appropriate fractions gave **2** as an oil (2.0 g, 5.37 mmol, 71%). MS (ESI): $m/z$ 373 (M$^+$+H). $^1$H NMR (400 MHz, CDCl$_3$) δ 7.33–7.22 (m, 4H), 6.94–6.82 (m, 4H), 5.89 (ddt, $J = 17.3, 10.5, 5.5$ Hz, 1H), 5.26 (dq, $J = 17.3, 1.7$ Hz, 1H), 5.21–5.12 (m, 1H), 4.62 (s, 2H), 4.47 (s, 2H), 4.01–3.98 (m, 2H), 3.81 (s, 3H), 3.80 (s, 3H), 3.75 (dt, $J = 5.6, 4.7$ Hz, 1H), 3.65–3.47 (m, 4H).

**(S)−2,3-bis((4-methoxybenzyl)oxy)propan-1-ol (3).** A mixture of **2** (370.0 mg, 0.99 mmol), PdCl$_2$ (17.6 mg, 0.99 mmol), and CuCl (98.0 mg, 0.99 mmol) in DMF/water (10/1, v/v, 11 mL) was stirred overnight under nitrogen. The mixture was then diluted with diethyl ether (10 mL) and filtered over Celite. The filtrate was washed with water (5 mL), and the layers were separated. The organic phase was dried over MgSO$_4$, filtered, and concentrated in vacuo. The residual oil was purified by silica gel column chromatography (eluent: ethyl acetate/hexanes) to afford **3** (144.0 mg, 0.43 mmol, 44%); MS (ESI): $m/z$ 333 (M$^+$+H). $^1$H NMR (400 MHz, CDCl$_3$) δ 7.27–7.23 (m, 4H), 6.94–6.81 (m,

4H), 4.66–4.51 (m, 2H), 4.47 (d, $J$ = 2.1 Hz, 2H), 3.81 (s, 3H), 3.80 (s, 3H), 3.75–3.53 (m, 5H).

**((2R,3R,4S,5R,6S)–3,4,5-tris(benzyloxy)–6-((S)–2,3-bis((4-methoxybenzyl)oxy)propoxy) tetrahydro-2H-pyran-2-yl)methyl acetate (5).** Freshly made **4** ((2R,3R,4S,5R,6R)–3,4,5-tris(benzyloxy)–6-iodotetrahydro-2H-pyran-2-yl)methyl acetate, 1.53 g, 2.55 mmol)[49] was dissolved in dry $CH_2Cl_2$ (25 mL) and cannulated into a stirred mixture of **3** (470.0 mg, 1.41 mmol), tetrabutylammonium iodide (TBAI, 1.83 g, 4.96 mmol), 2,4,6-tri-tert-butylpyrimidine (969.0 mg, 3.90 mmol), and freshly activated 4 Å molecular sieves in dry $CH_2Cl_2$ (20 mL). The reaction was stirred for 4 days at room temperature under $N_2$. The mixture was diluted with Ethyl acetate (20 ml), quenched with 1 M $Na_2S_2O_3$, and then filtered through Celite. The filtrate was washed with water and brine, dried over $MgSO_4$, and concentrated under reduced pressure. The residue was diluted with $Et_2O$ (20 ml), and the remaining TBAI was excluded by filtration. The filtrate was concentrated, and the residue was purified by flash chromatography (eluent: ethyl acetate/hexanes) to give **5** (687.0 mg, 0.85 mmol, 60%). MS (ESI): $m/z$ 807 ($M^+$+H). $^1$H NMR (400 MHz, $CDCl_3$) δ 7.36–7.13 (m, 19H), 6.86–6.70 (m, 4H), 4.96 (d, $J$ = 10.8 Hz, 1H), 4.86–4.73 (m, 3H), 4.70–4.49 (m, 5H), 4.44–4.39 (m, 2H), 4.20–4.09 (m, 2H), 3.96 (dd, $J$ = 9.6, 8.9 Hz, 1H), 3.82–3.76 (m, 1H), 3.76–3.73 (m, 1H), 3.72 (s, 3H), 3.71 (s, 3H), 3.57–3.40 (m, 5H), 1.94 (s, 3H).

**((2R,3R,4S,5R,6S)–3,4,5-tris(benzyloxy)–6-((S)–2,3-bis((4-methoxybenzyl)oxy)propoxy) tetrahydro-2H-pyran-2-yl)methanol (6).** NaOMe in MeOH (25% wt., 350 μL) was added to a stirred solution of **5** (400.0 mg, 0.49 mmol) in $CH_2Cl_2$ (5 mL). The mixture was stirred for half an hour and then neutralized with Amberlite 120R ($H^+$ form) resin. The mixture was filtered, eluent concentrated, and the residue was purified by flash chromatography (eluent: ethyl acetate/hexanes) to obtain **6** (350.0 mg, 0.46 mmol, 92%). MS (ESI): $m/z$ 787 ($M^+$+Na).$^1$H NMR (400 MHz, $CDCl_3$) δ 7.39–7.18 (m, 20H), 6.91–6.72 (m, 4H), 4.98 (d, $J$ = 10.9 Hz, 1H), 4.93–4.78 (m, 3H), 4.77–4.57 (m, 5H), 4.45 (s, 2H), 3.99 (t, $J$ = 9.3 Hz, 1H), 3.80 (dt, $J$ = 5.6, 2.6 Hz, 2H), 3.78 (s, 3H), 3.77 (s, 3H), 3.70–3.64 (m, 2H), 3.59–3.48 (m, 5H).

**(2S,3S,4S,5R,6S)–3,4,5-tris(benzyloxy)–6-((S)–2,3-bis((4-methoxybenzyl)oxy)propoxy) tetrahydro-2H-pyran-2-carboxylic acid (7).** **6** (950.0 mg, 1.24 mmol), 2,2,6,6-tetramethyl-1-(l1-oxidanyl)piperidine (TEMPO, 100.0 mg, 0.64 mmol), and (Diacetoxyiodo)benzene (BAIB, 2068.0 mg, 6.42 mmol) in a mixture of $CH_2Cl_2$/$H_2O$ (2:1, 12 mL) was stirred at room temperature for 3 h. The reaction was quenched with 0.25 M $Na_2S_2O_3$, and the resulting biphase was extracted with ethyl acetate (15 ml). The organic layer was washed with water, dried over $MgSO_4$, and concentrated under reduced pressure. The residue was purified by flash chromatography (eluent: ethyl acetate/hexanes) to obtain **7** (850.0 mg, 1.09 mmol, 88%). MS (ESI): $m/z$ 777 ($M^+$-H) $^1$H NMR (400 MHz, $CDCl_3$) δ 7.39–7.16 (m, 19H), 6.87–6.79 (m, 4H), 4.97 (d, $J$ = 10.9 Hz, 1H), 4.88–4.79 (m, 3H), 4.74–4.66 (m, 2H), 4.65–4.58 (m, 3H), 4.46 (s, 2H), 4.28 (d, $J$ = 10.1 Hz, 1H), 4.05–3.94 (m, 1H), 3.88–3.78 (m, 2H), 3.77 (s, 3H), 3.76 (s, 3H), 3.70 (dd, $J$ = 10.2, 8.8 Hz, 1H), 3.62–3.54 (m, 4H).

**benzyl (2S,3S,4S,5R,6S)–3,4,5-tris(benzyloxy)–6-((S)–2,3-bis((4-methoxybenzyl)oxy) propoxy)tetrahydro-2H-pyran-2-carboxylate (8).** Benzyl alcohol (138.0 mg, 1.28 mmol) was added to a stirred mixture of **7** (663.0 mg, 0.85 mmol), 2-(1H-benzotriazol-1-yl)–1,1,3,3-tetramethyluronium hexafluorophosphate (HBTU, 646.0 mg, 1.70 mmol), N,N-Diisopropylethylamine (DIPEA, 275.0 mg, 2.13 mmol), and N,N-dimethylpyridin-4-amine (DMAP, 41.6 mg, 0.34 mmol) in $CH_2Cl_2$ (10 mL). The mixture was stirred overnight at room temperature and then quenched with water. The organic layer was then concentrated under reduced pressure, and the residue was purified by

flash chromatography (eluent: ethyl acetate/hexanes) to obtain **8** (250.0 mg, 0.29 mmol, 34%). MS (ESI): $m/z$ 892 ($M^+$+Na). $^1$H NMR (400 MHz, $CDCl_3$) δ 7.44–7.07 (m, 24H), 6.87–6.75 (m, 4H), 5.14 (s, 2H), 4.97–4.84 (m, 2H), 4.82–4.68 (m, 3H), 4.65–4.55 (m, 3H), 4.44 (d, $J$ = 9.1 Hz, 3H), 4.36–4.27 (m, 1H), 3.98 (t, $J$ = 9.3 Hz, 1H), 3.87–3.77 (m, 3H), 3.76 (s, 3H), 3.75 (s, 3H), 3.75–3.70 (m, 1H), 3.62–3.52 (m, 3H), 2.17 (s, 5H).

**benzyl (2S,3S,4S,5R,6S)–3,4,5-tris(benzyloxy)–6-((S)–2,3-dihydroxypropoxy)tetrahydro-2H-pyran-2-carboxylate (9).** Ceric ammonium nitrate (CAN, 1794.0 mg, 3.27 mmol) was added to a stirred solution of **8** (948.0 mg, 1.09 mmol) in MeCN/$H_2O$ (11:1, 12 mL) at room temperature. The mixture was stirred at room temperature for 2 h, then diluted with ethyl acetate (20 ml) and $H_2O$ to create a biphase. The organic layer was washed with water and brine, dried over $MgSO_4$, and concentrated under reduced pressure. The residue to obtain crude **9** (300.0 mg, 0.48 mmol, 44%). MS (ESI): $m/z$ 629 ($M^+$+H). $^1$H NMR (400 MHz, Chloroform-$d$) δ 7.30–7.26 (m, 5H), 7.26–7.22 (m, 6H), 7.22–7.18 (m, 7H), 7.09–7.04 (m, 2H), 5.16–5.04 (m, 2H), 4.85 (d, $J$ = 10.92 Hz, 1H), 4.79–4.71 (m, 2H), 4.71–4.61 (m, 2H), 4.57 (d, $J$ = 11.80 Hz, 1H), 4.39 (d, $J$ = 10.74 Hz, 1H), 4.23 (d, $J$ = 9.92 Hz, 1H), 3.90 (t, $J$ = 9.35 Hz, 1H), 3.87–3.74 (m, 2H), 3.72–3.64 (m, 2H), 3.54 (dt, $J$ = 9.62, 5.04 Hz, 2H), 3.41–3.32 (m, 1H).

**(2S,3S,4S,5R,6S)–6-((R)–2,3-bis(palmitoyloxy)propoxy)–3,4,5-trihydroxytetrahydro-2H-pyran-2-carboxylic acid (GlcA DAG 16:0/16:0, (10)).** Palmitic acid (53.0 mg, 0.207 mmol), 4-(6-cyano-2-methyl-7-oxo-4,8-dioxa-2,5-diazadec-5-en-3-ylidene)morpholin-4-ium hexafluorophosphate(V) (COMU, 89.0 mg, 0.21 mmol), and DMAP (38.0 mg, 0.31 mmol) was added to a stirred solution of crude **9** (65.0 mg, 0.10 mmol) in DMF (5 mL) and the mixture stirred under $N_2$ for 24 h. A second equal portion of palmitic acid, COMU, and DMAP was then added as above, and the mixture was stirred for a further 24 h to drive the reaction to completion. The mixture was then diluted with ethyl acetate (10 ml) and washed with saturated $NaHCO_3$, water, and brine. Then dried over MgSO4 and concentrated under reduced pressure. The residue was purified by flash chromatography (eluent: ethyl acetate/hexanes) to obtain the intermediate ester, which was taken directly forward to deprotection. A mixture of Pd(OH)$_2$/C (20%, 10.0 mg), intermediate ester (10.0 mg, 9.05 μmol) in MeOH/THF (3:2, 5 mL) containing acetic acid (40 μL) was stirred under a $H_2$ atmosphere for 2 h. Then degassed and filtered through a layer of Celite and the filtrate was concentrated under reduced pressure. Final purification by silica gel chromatography of the residue (eluent: $CHCl_3$/MeOH) gave the product **10** (GlcA DAG 16:0/16:0, 6.0 mg, 8.05 μmol, 8%). $^1$H NMR (500 MHz, DMSO-$d_6$) δ 5.11 (td, $J$ = 8.17, 5.31 Hz, 1H), 4.94 (d, $J$ = 5.11 Hz, 1H), 4.83 (d, $J$ = 6.55 Hz, 1H), 4.70 (t, $J$ = 2.89 Hz, 1H), 4.38–4.27 (m, 1H), 4.20–4.13 (m, 1H), 3.86–3.75 (m, 1H), 3.72–3.66 (m, 1H), 3.56–3.51 (m, 1H), 3.38 (q, $J$ = 7.04 Hz, 2H), 3.23 (ddd, $J$ = 9.71, 6.35, 3.72 Hz, 1H), 2.27 (td, $J$ = 7.31, 2.96 Hz, 4H), 1.23 (s, 52H), 0.85 (t, $J$ = 6.82 Hz, 6H). HRMS-ESI calculated for $C_{41}H_{75}O_{11}$ [M-H]$^-$ 743.53094, found: 743.53131.

**Glycerophosphocholine sole phosphate source experiments**
Pi-free and 25 mM Pi-replete modified media were constituted based upon Middlebrook 7H9 media. The base for both modified media had the following composition:

Per 900 mL base:
Ammonium sulphate 0.5 g
L-Glutamic Acid 0.5 g
Sodium citrate 0.1 g
Ferric ammonium citrate 0.04 g
$MgSO_4$ 0.05 g
KCl 0.547 g
0.1 M pyridoxine 59 μL

0.1 M biotin 20.8 µL
0.1 M CaCl$_2$ 45 µL
0.1 M ZnSO$_4$ 34.7 µL
0.1 M CuSO$_4$ 62.5 µL
1:2 glycerol (50%) 4 mL
MOPS buffer 22 g
dH$_2$O 896 mL

From these 900 mL aliquots, to prepare Pi-free media, 1.46 g NaCl was added.

To make Pi-replete (25 mM) media, 3.0 g of NaH$_2$PO$_4$ was instead added.

Tyloxapol was added to each to a final concentration of 0.05%, and pH was adjusted to pH 6.6 using concentrated NaOH. Finally, 100 mL of 10× ADC growth supplement was added to make up a final volume of 1 L. Media were filter sterilised through a 0.22 µm membrane.

Bacteria were grown in 7H9(c) until OD$_{600}$ = 1. These were used to inoculate pre-conditioning cultures, of 24 mL volume in Falcon$^{TM}$ conical centrifuge tubes, at a starting OD$_{600}$ = 0.06. For the Pi-free arm this stage was Pi-free medium, for the high-Pi arm this was Pi-replete (25 mM) medium. These were incubated at 37 °C with continuous rotation for 72-h. Bacteria were then transferred into 100 mL of fresh media of the same Pi concentration as in the pre-culture stage, at a starting OD$_{600}$ = 0.06, in roller bottles, and growth curves commenced.

Once the Pi-free cultures had reached growth stasis, defined as 48-h at a static OD$_{600}$, bacteria from these cultures were transferred into fresh Pi-free medium (negative control), fresh Pi-replete (25 mM) medium (positive control) or into fresh Pi-free medium to which GroPCho had been added to a final concentration of 25 mM. These cultures were all of 24 mL volume in 50 mL Falcon$^{TM}$ tubes. These were incubated at 37 °C with continuous rotation.

Each condition/strain was performed in duplicate cultures and the experiment was performed twice.

### Electron Microscopy analysis of Mtb cell envelope

**Sample preparation.** Mtb WT samples were grown to stationary phase in zero phosphate media and 25 mM phosphate media as detailed above. At this point samples were centrifuged at 3000 × $g$, supernatants removed and fixed by adding 2 ml of 4% PFA in 200 mM HEPES (pH 7.4) buffer to each bacteria pellet. Samples were incubated at room temperature for 15 min and then replaced with 4% PFA in 100 mM HEPES overnight at 4 °C before further processing.

**Cryo-EM grid preparation, data acquisition and image processing.** The fixed bacterial cells were seeded on Quantifoil grids (carbon R 1.2/1.3 on Cu 300 mesh, Quantifoil Micro Tools GmbH, Germany). Grids were loaded into the chamber of Vitrobot Mark IV (Thermo Fisher Scientific, USA) equilibrated at 22 °C and 94% humidity, blotted for 12 s and vitrified in liquid ethane.

Cryo-EM images were collected on a Talos Arctica microscope (Thermo Fisher Scientific, USA) operating at 200 kV using Tomography software (v5.21, Thermo Fisher Scientific, USA) at magnification corresponding to 3.25 Å pixel size and a nominal defocus range −8 to −4 µm. Images were recorded using a Falcon III camera (Thermo Fisher Scientific, USA) in linear mode with a total dose of 53.67 electrons per Å2 fractionated over 50 frames (dose rate 56.75 e-/Å2/s).

The movies were imported into Relion (v 5.0.0)[75], followed by beam-induced motion correction using Relion's implementation of the UCSF motioncor2 program for whole-frame movie alignment, and CTF estimation using CTFFIND v4.1[76].

**Image analysis and quantification.** Micrographs were opened using Fiji/Image J (Version:2.16.0/1.54p). Images were converted to.tif

format. Adjustments were made to the brightness and contrast for visualization without altering raw data used for the quantifications. Regions of interested were randomly selected using the rectangle selection tool and covered the distance from the MOM to the PM. Pixel values were measured using the plot profile function. MOM and PM were characterized by morphological features and measurements taken from peak to peak, as per ref. 77. Data values represent the distance in scaled units (nm) between the two peaks. Data was analyzed using GraphPad prism (Version 10.1.1).

### Mouse infection experiments

All infection studies were approved by the Francis Crick Institute Ethics Committee and performed under a UK Home Office approved Animal License (P4D8F6075). 6 to 8-week-old C57BL/6J female mice were used for all experiments, these were bred and housed under specific pathogen-free conditions in the Biological Research Facility at the Francis Crick Institute.

All mice were maintained in Biological Safety Level 3 cages, at 22 ± 2 °C with a relative humidity of 55 ± 10% and 12 h light/dark cycle. Mice were maintained with ad libitum access to food and water. Procedures involving mice were performed in strict accordance with the United Kingdom Animals (Scientific Procedures) Act 1986 and the Institute's policies on the Care, Welfare and Treatment of Animals.

The mice were exposed to low-dose aerosolised TB infection in an infection chamber using a modified Glas-Col nebuliser system (Glas-Col, Terre Haute, USA). WT H37Rv and deletion strains were grown to mid log-phase in Middlebrook 7H9 media containing ADC and then diluted to produce an infection dose of ~100 CFU/mouse lung. Immediately after infection lungs were removed from 5 mice/infection group to assess the infection load CFU. Thereafter, mycobacterial CFU are assessed in the lungs of 5 mice/infection group at set time-points after infection: 30, 60, 90 and 120 days.

For CFU analysis, whole lungs were homogenised in tubes containing a solution of saline/0.01% digitonin (Merck, CAS number 11024-24-1) with sterile 3 mm glass beads (Merck) using a FastPrep-24 homogenisation system (MP-Biomedicals Inc.). Lung homogenates were serially diluted tenfold, and duplicate samples plated onto Middlebrook 7H11 agar plates containing OADC supplement. Plates were incubated at 37 °C and assessed from day 14 post plating.

### PDIM extraction and thin layer chromatography (TLC)

20 mL of bacterial culture for each strain (Middlebrook 7H9(c) media at OD$_{600}$ = 0.2) was centrifuged at 2000 × $g$ for 5 min. Supernatant was discarded and the pellet resuspended in 1 mL of phosphate buffered saline and transferred into a 2 mL screw-top cap. These were heated for 2 h at 92 °C in a water bath to ensure bacterial killing. Samples were centrifuged at 21,100 × $g$ and pellets washed in 1 mL Milli-Q water 3 times. The resulting pellet was resuspended in 1 mL methanol and transferred into a glass tube. 500 µL chloroform was added to generate 2:1 (v/v) methanol: chloroform solution. This underwent mixing overnight on an automated rotator at room temperature. Next day, samples were centrifuged at 100 × $g$ for 10 min, and supernatants transferred into new glass vials and stored at room temperature. The pellets were resuspended in 1 mL methanol: 1 mL chloroform, and mixed for 5 h on the automated rotator and room temperature. Samples were then centrifuged for 5 min at 100 × $g$, and supernatant was taken and combined with the previous supernatant. These pooled supernatant samples were dried under nitrogen stream, then resuspended in 100 µL of dichloromethane and vortexed vigorously. 10 µL per sample was spotted onto silica gel 60 F$_{254}$ TLC plates (Sigma-Aldrich) using a glass syringe. PDIM standard dissolved in dichloromethane was ran as a control. Purified PDIM was provided by American Type Culture Collection, (ATCC), Virginia USA. Plates were treated twice in petroleumether: diethylether (v/v) 9:1.

After air drying, plates were stained with 5% phosphomolybdic acid in ethanol (solution at 4 °C) and developed by drying with a heat gun. The TLC was performed twice.

## Statistics and reproducibility

For volcano plots for visualizing metabolomic and lipidomic datasets, peaks were automatically integrated on Progenesis QI v3.0 and the values normalized to total ion abundance. A filter of co-efficient of variation (CV) < 30 was applied in Progenesis, and only the features meeting this filter were exported to Microsoft Excel. Mean abundance was then calculated across the replicates within each strain/condition. These mean values were used to calculate fold changes, which are plotted as the $\log_2$ transformation on the $x$-axis. To facilitate fold-change calculations, when a feature was below the limit of detection, a missing value of 1000 was assigned.

Two tailed Student's $t$-test was used to compare means between groups, and were calculated using the individual values across all replicates within each group. The resulting $p$-values were then adjusted for multiple comparisons using a false discovery rate approach with a desired FDR of 5%, calculated using GraphPad Prism (Version 10.1.1). The $-\log_{10}$ transformation of these values are plotted on the $y$-axis. All volcano plots were generated using OriginPro 2025. For the negative ion-mode data in the metabolomics dataset in Fig. 1a, all features that were significantly altered in the $\Delta glpQ1$ strain versus the WT were further analyzed in Freestyle 1.8 SP2, including the manual integration of the AUC, and these values were used, after normalization to total ion abundance, to generate the volcano plot.

For specific later analyzes, following confident feature annotation based on accurate mass, retention time and HCD data, species were no longer excluded if they failed the CV < 30 filter. Therefore more species may have been included in these analyzes than appear in the corresponding volcano plots. Generally, 5 or more replicate cultures were analyzed per strain/condition, and exact numbers are detailed in the figure legends. No statistical method was used to predetermine sample size, instead, for LC-MS experiments 5–6 replicates were performed, as this is our typical practice and generates robust data. Generally, the experiments were not randomized, except that the LC-MS samples were run on the apparatus in random order, to reduce the chance of technical variations biasing the outcome. The Investigators were not blinded to allocation during experiments and outcome assessment. No data were excluded from the analyzes with the following exceptions: LC-MS samples were excluded if their normalization factor (normalization to total ion intensity) was greater than twofold. For volcano plots, as stated above, features failing the CV < 30 statistical filter were not included in the plots, but may have been further interrogated. For the EIC overlay in Fig. 2b, as described in the figure legend, 2 outliers were not included in the overlay plot, but their values are included in the calculations of statistical significance for the differences between the strains, to prevent bias.

## Ethical approval

All studies complied with all relevant ethical regulations, specifically the animal studies were approved by the Francis Crick Institute Ethics Committee and performed under a UK Home Office approved Animal License (P4D8F6075).

## Reporting summary

Further information on research design is available in the Nature Portfolio Reporting Summary linked to this article.

## Data availability

The LC-MS datasets generated and analysed during the current study are available in the Francis Crick Institute Figshare platform, at https://doi.org/10.25418/crick.29646266. Source data are provided with this paper.

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

## Acknowledgements

This work was supported by the Francis Crick Institute which receives its core funding from Cancer Research UK (FC001060 and CC2081), the UK Medical Research Council (FC001060 and CC2081), and the Wellcome Trust (FC001060 and CC2081). This work was partially funded by a Wellcome Trust New Investigator Award (104785/B/14/Z) to L.P.S.d.C. and by relocation funds from the Herbert Wertheim UF Scripps Institute for Biomedical Innovation and Technology to L.P.S.d.C. R.E.L, and J.L. received funding from ALSAC, St Jude Children's Research Hospital. We thank Natasha Lukoyanova and the Structural Biology Science Technology Platform at the Francis Crick Institute for the acquisition of the cryo-electron microscopy images and for advice relating to these experiments.

## Author contributions

R.M.G. devised the experiments, conducted the experiments and wrote the manuscript. L.P.S.d.C devised the project, devised the experiments and edited the manuscript. D.M.H. A.G-G., and A.A. generated the mutant strains of Mtb. M.S.d.S., A.G-G., and J.M. designed and performed the LC-MS. R.E.L. and J.L. devised and performed the chemical synthesis. M.G.G. was involved in the design of the mouse and electron microscopy experiments. A.R. carried out the mouse experiments under licence. J.O.C. prepared bacterial samples for electron microscopy experiments and A.F. performed the Mtb microscopy experiments and analysis.

## Funding

## Competing interests

The authors declare no competing interests.
