## [Transparent Peer Review file · Nature Communications]

Mycobacterium tuberculosis overcomes phosphate starvation by extensively remodelling its lipidome with phosphorus-free lipids.

Corresponding Author: Dr Luiz Pedro de Carvalho

Version 0:

Reviewer comments:

Reviewer #1

(Remarks to the Author)

Gray et al studied the lipidome of Mtb during Pi restriction, showing that under these conditions it is extensively remodelled. They show that phospholipids are replaced with different classes of phosphorus-free lipids, including some novel ones. They raise the hypothesis that this alteration of the Mtb lipidome in response to Pi restriction is an important strategy evolved to evade the conditions faced during infection of alveolar macrophages. Although interesting, they do not provide evidence that such might indeed be the case. Of importance, they show that culturing Mtb in standard media or Pi-free media yields different results, raising awareness for the use of liquid cultures to test antibiotics for example.

They start by generating an in-frame clean genetic deletion of the enzyme GlpQ1 in H37Rv and position this enzyme as a glycerophosphodiesterase of phospholipids in Mtb. They also constructed a complemented strain and show a phenotypic reversion of about 50% - in particular the reversion of the growth phenotype of the complemented strain in figure 2d seems almost absent. In this sense, the statement "Mtb can access host produced phospholipid polar heads, in particular GroPCho, as a source of phosphate, in a GlpQ1-dependent manner" is not fully convincing. In addition to the metabolomic/lipidomic analysis they performed with the glpQ1 mutant, they infected mice with this strain and compared it directly to the WT. Why the complemented strain was not included in these experiments is not clear. Also, they focused the in vivo experiment in time points post 30 days of infection, and only in levels of bacterial burden. Given that alveolar macrophages play a role in Mtb infections is much earlier, was there any difference at earlier time points? Was there any effect on tissue pathology, ie, beyond bacterial burdens? The two in vivo experiments performed yielded somehow different results, leaving the question as whether GlpQ1 is important for Mtb virulence or not unanswered. It would be important to have statistical analysis of the data in extended data 5a.

They then show that Mtb profoundly remodelled its lipidome when cultured in Pi-free medium (figure 3), however in a GplQ1-independent manner. This is a very interesting result. Are there any candidate enzymes involved in this? The authors consider that "Both mechanisms are likely important in macrophage infection (where Pi concentrations are below 10 μ M) during the pathogenesis of pulmonary TB." Could they probe this hypothesis for example in an in vitro system?

The authors need to check the text and figures carefully. There are panels in extended data are not referred to in the text (e.g. extended data 5b); others that are referred to in the text but not labelled accordingly in the figure (e.g. figure 3c: i, ii, iii are not labelled in the figure); and in several cases the panels are not cited in order in the text (e.g. extended data 3c-e is cited before extended data 2c). Statistical analysis should be performed in extended data 5a. The sentence "immunologists and vaccinologists have not studied the immune response to physiologic, phosphate-poor Mtb" is not clear, as the immune response is mostly studied in vivo, where the Pi restriction and the lipidome remodelling should presumably occur.

Reviewer #2

(Remarks to the Author)

This is a very interesting and novel paper by Gray et al., where they show that Mycobacterium tuberculosis (Mtb) remodels its cell envelope lipids under phosphate (Pi) starvation by reducing the amount of phospholipids and increasing the

abundance of phosphorous-free lipids. In addition, authors show that Mtb metabolizes phospholipid polar heads from lipids associated to the host environment, more specifically from the pulmonary surfactant. Taking into consideration the real and physiological condition of Pi starvation that Mtb faces within alveolar macrophages during infection, authors emphasize the artificial way we typically grow Mtb in the lab, which includes media with supra physiological amounts of Pi. Therefore, authors recognize that this study could open new ways to explore pathways involved in uptake, synthesis or degradation these new lipid species to find novel drug targets or vaccine candidates.

To show this, authors use mycobacterial growth assays, lipidomics and metabolomics. Even though authors recognize that this capacity (remodeling lipids in the absence of Pi to phosphorous-free lipids) has been previously described in unrelated environmental and pathogenic bacteria along with plants or algae, the study is important for the field considering the relevance of lipids in mycobacterium physiology. However, there are some aspects that may need further clarification:

1. The most important limitation of this study is the use of glycerol as a sole carbon source. This reviewer wonders about the impact of more relevant carbon sources such as cholesterol or fatty acids may have in the lipid composition of Mtb in the absence of Pi. To properly dimension the importance of the growth medium that we can mimic the infecting niche, the carbon source should be taking into consideration.
2. Authors need to provide some more context on why they choose GlpQ1.
3. Considering the magnitude of the growth restriction provided by the absence of Pi, authors need to provide some data about bacterial viability.
4. Is it LAM also modified in the absence of Pi.
5. Could authors provide some ultrastructural features of mycobacteria grown in the absence of Pi?. This would help to assess the integrity of the cells.
6. There is very detailed and sophisticated mass spectrometry approach to define the lipid repertoire. However, do the authors find worthwhile to run a transcriptomic study on the glpQ1 mutant to investigate the potential compensatory mechanisms?
7. Please, clarify whether GroPCho is the lipid head of dipalmitoylphosphatidylcholine, phosphatidylcholine or both. (see line 111 page 5 and line 138, page 6).
8. It is not clear, considering the magnitude of growth, whether the access to GroPCho by Mtb reactivates its growth. Please, provide CFUs data.
9. There is a general lack of statistics through the text.
10. When discussing the potential attenuation of the glpQ1 mutant in the context of transposon studies or previous studies, authors should investigate the vulnerability of this gene in Mtb in the context of CRISPRi interference. This information is available. Even if the authors question such studies because of the growth medium, it should be noted in the study.
11. Looking at the in vivo infection study, it seems that lower infecting inoculum (exp 2) may reduce differences in bacterial numbers between the WT and the mutant; and having an infecting dose of 200 CFUs, this reviewer wonders whether even lower infecting dose may eliminate differences between WT and the mutant. The fact that this experiment does not include a complemented strain make the results not conclusive.
12. Another factor that may have not been evaluated in the study is the influence of tyloxapol. As detergent it is known to shave mycobacterial capsule and exert some influence on the levels of specific lipids, some of them analyzed in this study. Therefore, it would be ideal to show that tyloxapol, another artificial component of growth medium, is not affecting the overall results of this study.

Minor issues;

1. Please, add figure labels to main and extended figures.

Reviewer #3

(Remarks to the Author)

The manuscript is very well written, clear and contains no overstatements. The study is divided into two parts: 1) study of the role of the GlpQ1 enzyme as a glycerophosphodiesterase 2) description of the in vitro remodeling of the mycobacterial lipidome under Pi deprivation conditions.

The novelty of the results lies in the characterization of a new enzyme whose functionality contributes to bacillus pathogeny, and in the identification of new lipids whose production is part of the bacillus response to Pi deprivation in vitro. Results are of interest to the field. However, the below points should be addressed to reinforce the solidity of the results of this study and the quality of the manuscript.

- 1- In a general way, the precise list of acylforms, detected m/z, mass accuracy as compared to calculated m/z and RT of

each acylform used should be provided to definitely support annotations. The level of each phospholipid class is presented as a mean/summed abundance of several species. It would be appreciable if the mass spectrometry signal of the acylform family is shown for the different lipid classes (in supplemental) and it can be recommended that only the characteristic and main acylforms were summed for the robustness of the results. For example, PI 42:0 species is strikingly much more altered than other PI although it is likely not a major acylform of the PI family. In addition the meaning of carbon number in PL or PIM annotations should be explained (total carbons or alkyl chains sum ?).

2-Important accumulation of the glycerophosphothreonine head annotated in Fig 1a for the GlpQ1 mutant is not commented while it is yet unclear whether phosphatidylthreonine is even produced by mycobacteria. Figure 1a : Orange and blue color codes of are not detailed in the legend.

3-In Figure 1c, phospholipid level changes should be shown as absolute fold change as in extended Figure 2b rather than by a heat map, which does not provide precise fold change information to the reader. Indeed, it is difficult to be convinced based on colors only that PE is unchanged and PG decreased. Red color for PE level suggests an increase in the mutant while its degradation product GroPEth is too. If fold changes are subtle, it's all the more reason to present absolute values rather than a heat map.

4-Analysis of GroPCho: this lipid polar head is detected in positive mode in Figure 2b and in negative mode in extended Figure 2a. It should be also annotated in the plot Figure 1a as it is said to be enriched in the GlpQ1 mutant. In figure 2b the screened m/z for the GroPCho head should be indicated and in 2c, the fragments in the MS2 spectrum should be interpreted with structures.

5-Authors claimed that PC can be used by mycobacteria as a source of Pi thank to the activity of GlpQ1. Hence, PC facilitate growth in a Pi free medium without being required even for the mutant strain. The growth curves of the different strains are not very different, particularly at short time points (statistical data not available). Analysis of the GroPCho level in the WT and mutant strains grown in this PC supplemented Pi-free medium would confirm the bacillus use of exogenous PC.

6-What is the point of extended figure 5b? These results are not mentioned in the main text. Extended figure 6c: "Total area under curve » columns are mislabeled.

7- The method section suffers from a lack of details in terms of data processing and bioinformatics tools, making it impossible to reproduce the analyses as it stands. To give just a few examples: Files formats, Software and their version, parameters for data processing (s/n, peak peaking parameters, RT and m/z tolerance parameters for peak alignment...), annotation method (manual or automated, ppm tolerance for annotation). In addition, raw data and extracted datasets must be made available and usable.

Version 1:

Reviewer comments:

Reviewer #1

(Remarks to the Author)

The authors have addressed the main points raised before, by adding to the manuscript experiments performed with a second complemented strain where the expression of *glpQ1* is under a strong promoter, and by improving their in vivo (mouse) data through the inclusion of the complemented strain(s) and statistical analyses. They have also adjusted their discussion to better reflect the findings.

I have no further comments.

Reviewer #2

(Remarks to the Author)

This reviewer acknowledges the great effort authors have made to address the suggestions and comments. This reviewer will only comment of those points that has not fully been addressed by its perspective, considering the rest of comments fully addressed and closed.

1. The most important limitation of this study is the use of glycerol as a sole carbon source. This reviewer wonders about the impact of more relevant carbon sources such as cholesterol or fatty acids may have in the lipid composition of *Mtb* in the absence of Pi. To properly dimension the importance of the growth medium that we can mimic the infecting niche, the carbon source should be taking into consideration.

Authors Response: This is of course an important consideration and an excellent suggestion. However, the focus of this paper is on phosphate physiology rather than carbon physiology. We have previously published regarding many aspects of *Mtb* carbon metabolism [12], and nitrogen metabolism [13]. Furthermore, although the major carbon source in these studies is glycerol, which is widely used in studies of *Mtb* metabolism but is of course not relevant in infection, we also know from this study that the albumin-dextrose-catalase growth supplement used also contains multiple classes of phospholipid and

fatty acids. To dissect the simultaneous contribution of available carbon, phosphate and indeed nitrogen nutrients available to Mtb upon the lipidome it displays would be a very large and complex work.

Finally, by using glycerol in these studies present in excess, whilst varying phosphate, allows us to determine that the effects seen in the phospholipids/ lipid-heads are not due to a carbon source effect.

Reviewer Response: This is not convincing. This reviewer was just suggesting checking the growth in the present of relevant carbon sources like fatty acids in controlled experiments using defined minimal medium.

5. Could authors provide some ultrastructural features of mycobacteria grown in the absence of Pi?. This would help to assess the integrity of the cells.

Authors Response: Following this suggestion with have now performed cryoEM of the WT bacteria grown in 25mM phosphate and in 0mM for 2 weeks. These results are now shown in the main manuscript in figure 4. This reveals that phosphate starved bacteria have a significant increase in the thickness of their cell envelope, as well as exhibiting a greater variance in envelope thickness, compared to bacteria grown in phosphate replete culture. This is an exciting finding, as it reflects the lipid remodelling characterised by our LC-MS studies. Please, see lines 310-317 in the new manuscript.

Reviewer Response: Authors need to provide electron micrographs of Mtb grown at 25 mM Phosphate and include it in Figure 4.

6. There is very detailed and sophisticated mass spectrometry approach to define the lipid repertoire. However, do the authors find worthwhile to run a transcriptomic study on the glpQ1 mutant to investigate the potential compensatory mechanisms?

Authors Response: Agree this would potentially be insightful, however transcriptional analysis is not a strength of our laboratory and has not been performed in this instance. Furthermore, the regulation of enzymatic processes is not solely controlled at the level of transcription, and so the compensatory mechanisms for the alterations in phospholipid and lipid head levels in the mutant might not be decipherable from a transcriptomic approach.

Reviewer Response: This is not a convincing response. Authors speculate about the potential transcriptional regulation of this enzyme and do not open the discussion about the potential compensation of deletion of glpQ1.

Reviewer #3

(Remarks to the Author)

The various comments have been addressed.

Minor needed edits:

Line 100: Cap missing to "higher"

In the new figure 1 it should be mentioned in the legend that 2 independent experiments are presented.

Supplementary File 2: mislabeling of PG variants m/z in the legend: "3766.5572" for the main acylform

Please see responses in blue. Line numbers refer to the line number in the revised manuscript when viewed in Microsoft Word with "All Markup" selected in the Review tab with "All Reviewers" selected under the "Show Markup" drop down list.

We have made additional alterations to the manuscript than those described here to bring it in line with Nature Communication's formatting instructions- these are all also indicated in the "All markup" view.

REVIEWER COMMENTS

Reviewer #1 (Remarks to the Author):

Gray et al studied the lipidome of Mtb during Pi restriction, showing that under these conditions it is extensively remodelled. They show that phospholipids are replaced with different classes of phosphorus-free lipids, including some novel ones. They raise the hypothesis that this alteration of the Mtb lipidome in response to Pi restriction is an important strategy evolved to evade the conditions faced during infection of alveolar macrophages. Although interesting, they do not provide evidence that such might indeed be the case. Of importance, they show that culturing Mtb in standard media or Pi-free media yields different results, raising awareness for the use of liquid cultures to test antibiotics for example.

They start by generating an in-frame clean genetic deletion of the enzyme GlpQ1 in H37Rv and position this enzyme as a glycerophosphodiesterase of phospholipids in Mtb. They also constructed a complemented strain and show a phenotypic reversion of about 50%- in particular the reversion of the growth phenotype of the complemented strain in figure 2d seems almost absent. In this sense, the statement "Mtb can access host produced phospholipid polar heads, in particular GroPCho, as a source of phosphate, in a GlpQ1-dependent manner" is not fully convincing.

Response:

We recognise the degree of complementation with the complement strain is incomplete; reaching full complementation in Mtb is frequently challenging to achieve. We have improved upon this by subsequently generating a second complement strain $\Delta glpQ1::glpQ1_Psmyc$ i.e. the gene of interest now under a stronger promotor. As now included in Supplementary Fig. 2b the metabolic phenotype reverts much more completely to WT than the original $\Delta glpQ1::glpQ1_Pimyc$. We did not initially intend to include more than one complement strain in the manuscript as this could cause confusion. However including $\Delta glpQ1::glpQ1_Psmyc$ data adds confidence to the phenotypes.

The growth phenotype experiment in Figure 2d suffers from the same issue of incomplete/moderate complementation seen with $\Delta glpQ1::glpQ1_Pimyc$. However, to repeat this experiment twice with the $\Delta glpQ1::glpQ1_Psmyc$ strain, which we considered, would be unfeasible as this would be two 6-week containment level 3 experiments and the costs associated are high (particularly the cost of glycerophosphocholine at sufficient quantities to achieve 25mM). The moderate reversion in the growth phenotype of the $\Delta glpQ1::glpQ1_Pimyc$ in 2d thus reflects the incomplete reversion of the metabolomic phenotype, which is regretful. We would point out that the reversion in the growth phenotype was slightly better in the repeat experiment (Supplementary Fig. 4). Furthermore, we are confident of our conclusion that GlpQ1 enhances growth on glycerophosphocholine as a sole phosphate source because:

- (i) Reproducibility: The results are highly reproducible, particularly the difference in the peak ODs reached for the WT and the knockout strains, across the two experiments shown, and also a smaller preliminary experiment (results not included in the paper as only involved 2 strains)
- (ii) Precedence: There is precedence in other organisms: Soli-dwelling pseudomonas species can grow in media with GroPCho as a sole phosphate source only when their secreted GlpQ1 enzymes are present, as shown by *glpQ* knockout [1]. In *S. cerevisiae*, growth on GroPCho as a sole-phosphate source is also dependent upon a glycerophosphodiesterase, Gde1p, as well as upon the associated glycerophosphodiester transporter, Git1p [2]. GroPCho was also shown to be a sufficient sole-source of phosphate *in vitro* for wildtype *Staphylococcus aureus* to actively grow, whereas only very slight growth was seen for a Δ *glpQ* *S.aureus* strain under the same conditions [3]. However, the finding that Mtb would also demonstrate a *glpQ1* dependent growth with GroPCho as a sole phosphate source was not a foregone conclusion, as Mtb has a distinct metabolism from other bacteria (and indeed yeasts) and also some GlpQ systems are known to definitely *not* play any role in phosphate acquisition- i.e. the GlpQ-GlpT system in *E.coli*. [4]. We now refer to the precedence in *S. aureus* in the discussion of our findings see line 375 onwards.
- (iii) Upregulation- the membrane bound transporter in Mtb which is demonstrated to import glycerophosphocholine, UgpABCE [5, 6], is transcriptionally induced in response to low phosphate (i.e. induced by the RegX3 phosphate response transcription factor) [7, 8].

In addition to the metabolomic/lipidomic analysis they performed with the *glpQ1* mutant, they infected mice with this strain and compared it directly to the WT. Why the complemented strain was not included in these experiments is not clear. Also, they focused the *in vivo* experiment in time points post 30 days of infection, and only in levels of bacterial burden. Given that alveolar macrophages play a role in Mtb infections is much earlier, was there any difference at earlier time points? Was there any effect on tissue pathology, ie, beyond bacterial burdens? The two *in vivo* experiments performed yielded somehow different results, leaving the question as whether GlpQ1 is important for Mtb virulence or not unanswered.

Response:

We now include three mouse experiments. The second and third contain the complements, with the *glpQ1* gene under the intermediate and strong promoters respectively. The first experiment was performed without a complement strain available.

The complements both fail to revert towards the WT level of virulence, so the question of whether the gene contributes to virulence remains unanswered. Complementation is frequently a problem in Mtb mouse experiments. Further- the performance of the WT is slightly inconsistent across experiments- 10^6 CFU/lung in the first experiment, $10^5/10^4$ in the repeats. However the consistent finding is that the knockout always shows attenuation vs the WT level, across the 3 repeats, by 1 or 2 log₁₀ units. We are transparent here- we suggest that the gene *may* contribute to virulence in the mouse model, but that the results are inconclusive. See lines 210-213.

Regarding earlier timepoints, this is a good suggestion. However this would need us to repeat the experiments using further mice. We have instead looked for a phenotype in macrophage

experiments. These however showed no difference in CFUs at 72 hours post infection between the strains. We would ideally prefer to look in macrophages at later timepoints- as we know that phosphate starvation in macrophages begins to activate phosphate starvation response genes in Mtb from day 3 post infection onwards. However, macrophage infection studies beyond 72 hours relying on CFUs are unreliable due to the technical issue of macrophages dying and releasing bacteria. Unfortunately, despite further experimentation we cannot confirm a role for GlpQ1 in Mtb virulence in mice or in macrophage culture, even though the mouse work is suggestive.

It would be important to have statistical analysis of the data in extended data 5a.

Response:

This is now included in supplementary fig. 5 for the three experiments. The differences seen between WT and KO are statistically significant in each of the three experiments.

They then show that Mtb profoundly remodelled its lipidome when cultured in Pi-free medium (figure 3), however in a GlpQ1-independent manner. This is a very interesting result. Are there any candidate enzymes involved in this?

Response:

We have identified candidate enzymes for the synthetic enzymes for two classes of the phosphorus-free “substitution” lipids: ornithine lipids and glucuronyl diacylglycerols. We did not want to include these in the manuscript as they have not been directly characterised, however we shall work to characterise them as a major future focus of the laboratory. Privately however; most bacterial species studied generate ornithine lipids using an OlsB, ornithine N-alpha acyltransferase, followed by OlsA, lyso-ornithine lipid O-acyltransferase, which are organised into a 2 gene *OLSB-OLSA* operon. Based on tBLASTn in H37Rv the candidate genes are at *rv3027c* and *rv3026c* respectively.

For the synthesis of glucuronyl diacylglycerols, the situation is complicated by the issue that bacterial glycosyltransferases are often promiscuous for the sugars that they will transfer, and if expressed in a heterologous organism can catalyse a different sugar transfer to the one they catalyse in their native organism. However, the candidate for the enzyme catalysing transfer of glucuronic acid to diacylglycerol in H37Rv is *rv0557*. This gene has proven hard to express in *E.coli* by other groups and the function of its product enzyme in Mtb is debated i.e. [9, 10], however it does appear to be a glycosyltransferase.

Intriguingly, all three of these candidate genes have been demonstrated in a genome wide screen to be transcriptionally upregulated in response to phosphate starvation [11], which supports their proposed roles in responding to phosphate starvation by increasing synthesis of these phosphorus-free substitution lipids.

The authors consider that “Both mechanisms are likely important in macrophage infection (where Pi concentrations are below 10 μ M) during the pathogenesis of pulmonary TB.” Could they probe this hypothesis for example in an in vitro system?

Response:

This is a great point, we are starting to pursue this but as with the experiments involving the candidate synthetic enzymes above, these experiments will likely take years to fulfil.

We have performed a 3-day macrophage infection since the first submission of the manuscript, however there were no differences in CFUs between the WT and *glpQ1* mutant strains. However, this is not unexpected, as in model infection systems phosphate response genes are thought to only be induced from day 3 post infection onwards. Longer infections of macrophages in culture suffer from loss of macrophage viability in vitro and bacilli being released extracellularly; this prevents the extension of the duration of the experiment for our purposes as we propose that Pi restriction is specific to the phagosomal compartment.

The authors need to check the text and figures carefully. There are panels in extended data are not referred to in the text (e.g. extended data 5b); others that are referred to in the text but not labelled accordingly in the figure (e.g. figure 3c: i, ii, iii are not labelled in the figure); and in several cases the panels are not cited in order in the text (e.g. extended data 3c-e is cited before extended data 2c).

Response:

Apologies for this. It is a result of making several versions of the manuscript prior to submission. This is now corrected- every part of all figures and supplementary figures is cited in the main text- as we have added a new main figure (figure 4) and moved some figures around, the labels are different from the original manuscript.

Statistical analysis should be performed in extended data 5a.

Response:

This is now displayed for all 3 experiments in supplementary figure 5 (a,b,c).

The sentence “immunologists and vaccinologists have not studied the immune response to physiologic, phosphate-poor Mtb” is not clear, as the immune response is mostly studied in vivo, where the Pi restriction and the lipidome remodelling should presumably occur.

Response:

We agree with this suggestion. This sentence has now been removed and changed to specifically discuss what is known about the phosphate-free lipids identified here in terms of their presentation on human CD1 molecules to CD1-restricted T cells. See discussion lines 409-414.

Reviewer #2 (Remarks to the Author):

This is a very interesting and novel paper by Gray et al., where they show that *Mycobacterium tuberculosis* (Mtb) remodels its cell envelope lipids under phosphate (Pi) starvation by reducing the amount of phospholipids and increasing the abundance of phosphorous-free lipids. In addition, authors show that Mtb metabolizes phospholipid polar heads from lipids associated to the host environment, more specifically from the pulmonary surfactant. Taking into consideration the real and physiological condition of Pi starvation that Mtb faces within alveolar macrophages during infection, authors emphasize the artificial way we typically grow Mtb in the lab, which includes media with supra physiological amounts of Pi. Therefore, authors recognize that this study could open new ways to explore pathways involved in uptake, synthesis or degradation these new lipid species to find novel drug targets or vaccine candidates.

To show this, authors use mycobacterial growth assays, lipidomics and metabolomics. Even though authors recognize that this capacity (remodeling lipids in the absence of Pi to phosphorous-free lipids) has been previously described in unrelated environmental and pathogenic bacteria along with plants or algae, the study is important for the field considering the relevance of lipids in mycobacterium physiology. However, there are some aspects that may need further clarification:

1. The most important limitation of this study is the use of glycerol as a sole carbon source. This reviewer wonders about the impact of more relevant carbon sources such as cholesterol or fatty acids may have in the lipid composition of Mtb in the absence of Pi. To properly dimension the importance of the growth medium that we can mimic the infecting niche, the carbon source should be taking into consideration.

Response: This is of course an important consideration and an excellent suggestion. However, the focus of this paper is on phosphate physiology rather than carbon physiology. We have previously published regarding many aspects of Mtb carbon metabolism [12], and nitrogen metabolism [13]. Furthermore, although the major carbon source in these studies is glycerol, which is widely used in studies of Mtb metabolism but is of course not relevant in infection, we also know from this study that the albumin-dextrose-catalase growth supplement used also contains multiple classes of phospholipid and fatty acids. To dissect the simultaneous contribution of available carbon, phosphate and indeed nitrogen nutrients available to Mtb upon the lipidome it displays would be a very large and complex work.

Finally, by using glycerol in these studies present in excess, whilst varying phosphate, allows us to determine that the effects seen in the phospholipids/ lipid-heads are not due to a carbon source effect.

2. Authors need to provide some more context on why they choose GlpQ1.

Response: We chose this enzyme initially as we were primarily interested in plasma membrane remodelling, and GlpQ1 is one enzyme upstream in the phospholipid catabolism pathway of the enzyme we had characterised in a previous study, Rv1692/ glycerol-3 phosphate phosphatase [14]. We have highlighted this more strongly at lines 70-77.

3. Considering the magnitude of the growth restriction provided by the absence of Pi, authors need to provide some data about bacterial viability.

Response: This has been studied previously: Rifat et al. [11] showed that in zero phosphate liquid media, Mtb which reaches a peak count of 10^8 CFU/mL at day 7, has decreased by 0.5 log₁₀ at day 14, but at days 21 and 28 remains steady at 10^7 CFU/mL. This is a remarkable degree of compensation to starvation. Also in our study, after starving Mtb of phosphate for c. 2 weeks, we saw that transferring to fresh media containing 25mM NaH₂PO₄ allowed reactivation of bacterial growth (Figure 2d). Therefore, our results appear to be very consistent with this earlier study, i.e. viability is largely preserved despite starvation.

4. Is it LAM also modified in the absence of Pi.

Response: Lipoarabinomannan, LAM, is an important lipoglycan in mycobacteria, “built” on a PIM anchor. As PIMs decrease, we might expect LAM levels to also fall. However, our LCMS based approach cannot be used to assay LAM, which is a much larger molecule (typically around 15 kDa) than the upper limit of masses that our mass spectrometry method is designed to detect (m/z range 200–3000 mass units).

5. Could authors provide some ultrastructural features of mycobacteria grown in the absence of P_i ? This would help to assess the integrity of the cells.

Response: Following this suggestion we have now performed cryoEM of the WT bacteria grown in 25mM phosphate and in 0mM for 2 weeks. These results are now shown in the main manuscript in figure 4. This reveals that phosphate starved bacteria have a significant increase in the thickness of their cell envelope, as well as exhibiting a greater variance in envelope thickness, compared to bacteria grown in phosphate replete culture. This is an exciting finding, as it reflects the lipid remodelling characterised by our LC-MS studies. Please, see lines 310-317 in the new manuscript.

6. There is very detailed and sophisticated mass spectrometry approach to define the lipid repertoire. However, do the authors find worthwhile to run a transcriptomic study on the *glpQ1* mutant to investigate the potential compensatory mechanisms?

Response:

Agree this would potentially be insightful, however transcriptional analysis is not a strength of our laboratory and has not been performed in this instance. Furthermore, the regulation of enzymatic processes is not solely controlled at the level of transcription, and so the compensatory mechanisms for the alterations in phospholipid and lipid head levels in the mutant might not be decipherable from a transcriptomic approach.

7. Please, clarify whether GroPCho is the lipid head of dipalmitoylphosphatidylcholine, phosphatidylcholine or both. (see line 111 page 5 and line 138, page 6).

Response:

Yes both. GroPCho is the lipid head of the phosphatidylcholine class of phospholipid. This class of phospholipid is of major importance in mammalian cells. One example is dipalmitoylphosphatidylcholine, DPPC, which is highly abundant in human pulmonary surfactant. DPPC is the acylform of phosphatidylcholine with two 16-carbon fully-saturated fatty acids (palmitic acids). This is clarified in the text now at line 191.

8. It is not clear, considering the magnitude of growth, whether the access to GroPCho by *Mtb* reactivates its growth. Please, provide CFUs data.

Response:

Unfortunately, we do not have CFU data available, and to repeat this experiment to contain two independent datasets with CFUs would be unfeasible, as there are very considerable reagent costs, and the time course of the experiment is 6 weeks (per experiment), and CFUs would add 4 or more

weeks on top. However, we are confident of our conclusion that GroPCho reactivates growth, because:

- (i) **Reproducibility:** The results are highly reproducible across the two experiments shown and also a smaller preliminary experiment (results not included in the paper as only involved 2 strains).
- (ii) **No growth in the negative control:** In both of these experiments, when starved, growth arrested Mtb was transferred into fresh 0mM Pi medium, no growth was seen (OD measurements remained at the inoculated level of OD₆₀₀= 0.1 or decreased). By contrast, reactivation of growth upon transfer, at the same timepoint, into 25mM GroPCho resulted in growth from OD 0.1 to OD 1- a logarithm.
- (iii) **Precedence:** There is precedence in other organisms: soli-dwelling pseudomonas species can grow in media with GroPCho as a sole phosphate source and only when their secreted GlpQ1 enzymes are present. [1]. *S. cerevisiae* can similarly grow on GroPCho as a sole-phosphate source and this is dependent upon a glycerophosphodiesterase, Gde1p ([15, 16]). GroPCho was shown to be a sufficient sole-source of phosphate (and carbon) *in vitro* for wildtype *S. aureus* to actively grow, whereas only very slight growth could be seen for a Δ glpQ *S. aureus* strain under the same conditions [3].

9. There is a general lack of statistics through the text.

Response: Many of the figures and supplementary figures carry asterisks to indicate statistical significance. However, we have taken on board the suggestion and added statistical analysis of the mouse data now shown in supplementary fig. 5, and the cryo-EM study of membrane thickness performed since the first submission of this manuscript displayed in figure 4 includes markers of statistical significance. In the LC-MS studies, by employing 5 or more replicate cultures per strain/condition, associated p values for differences between strains were typically very marked- i.e. $\times 10^{-4}$ to 10^{-15} in figure 1b.

10. When discussing the potential attenuation of the glpQ1 mutant in the context of transposon studies or previous studies, authors should investigate the vulnerability of this gene in Mtb in the context of CRISPRi interference. This information is available. Even if the authors question such studies because of the growth medium, it should be noted in the study.

Response: Thank you for this point. We have added this to our discussion upon the unclear contribution of GlpQ1 to Mtb survival in macrophage culture and *in vivo* (Lines 200-201). The gene is seen to be invulnerable/of very low vulnerability. We agree that inclusion of the outcome of the CRISPRi study complements the discussion of the results of the earlier transposon mutagenesis screens and adds an important dimension.

11. Looking at the *in vivo* infection study, it seems that lower infecting inoculum (exp 2) may reduce differences in bacterial numbers between the WT and the mutant; and having an infecting dose of 200 CFUs, this reviewer wonders whether even lower infecting dose may eliminate differences between WT and the mutant. The fact that this experiment does not include a complemented strain make the results not conclusive.

Response:

In response we have now included 3 independent mouse experiments in supplementary fig. 5c, and added statistical analyses. This does show that the KO, in all three experiments, is significantly attenuated versus the parent strain, by 1 to 2 log₁₀. These later studies include complement strains $\Delta glpQ1::glpQ1_Pimyc$ and $\Delta glpQ1::glpQ1_Psmyc$, however, as can be seen, neither complement shows successful complementation of the phenotype. Complementation in animal infection models in Mtb is often recognised to be difficult to achieve.

Experiment 1 was performed before we had a verified complement strain. In our first submission, we wished to avoid any confusion by mentioning two separate complement strains. However, we now present the work with both the complements, $\Delta glpQ1::glpQ1_Pimyc$, which was generated first, and $\Delta glpQ1::glpQ1_Psmyc$, which was subsequently generated in an attempt to improve upon the partial degree of reversion towards wild-type seen for several of the phenotypes with $\Delta glpQ1::glpQ1_Pimyc$. As seen in the revised manuscript here, $\Delta glpQ1::glpQ1_Psmyc$ exhibited improved reversion to wild-type levels of the lipid heads (Supplementary fig. 2b), but still fails to complement the mouse phenotype. There is also variation in the performance of the WT strain across the 3 mouse experiments. However, the KO always produces significantly lower CFU than the WT across all three mouse infection experiments, even when the performance of the WT is variable.

We consider our conclusions as consistent with the strength of evidence presented- we state that GlpQ1 deletion leads consistently to attenuation, but that due to the lack of demonstrable complementation in either of our complement strains, the *suggested* contribution of GlpQ1 to Mtb virulence remains inconclusive (see lines 202-213)

12. Another factor that may have not been evaluated in the study is the influence of tyloxapol. As detergent it is known to shave mycobacterial capsule and exert some influence on the levels of specific lipids, some of them analyzed in this study. Therefore, it would be ideal to show that tyloxapol, another artificial component of growth medium, is not affecting the overall results of this study.

Response: This is another excellent point. It is for this reason that in all of our lipidomic studies, we do not include tyloxapol (nor any other detergent) in the media, as it would introduce an important artefact as you point out.

Minor issues;

1. Please, add figure labels to main and extended figures.

Response: This is now done, apologies for the omission.

Reviewer #3 (Remarks to the Author):

The manuscript is very well written, clear and contains no overstatements. The study is divided into two parts: 1) study of the role of the GlpQ1 enzyme as a glycerophosphodiesterase 2) description of the in vitro remodeling of the mycobacterial lipidome under Pi deprivation conditions.

The novelty of the results lies in the characterization of a new enzyme whose functionality contributes to bacillus pathogeny, and in the identification of new lipids whose production is part of the bacillus response to Pi deprivation in vitro. Results are of interest to the field. However, the below points should be addressed to reinforce the solidity of the results of this study and the quality of the manuscript.

1- In a general way, the precise list of acylforms, detected m/z, mass accuracy as compared to calculated m/z and RT of each acylform used should be provided to definitely support annotations.

Response- This is now included: as Supplementary Tables 1-3.

The level of each phospholipid class is presented as a mean/summed abundance of several species. It would be appreciable if the mass spectrometry signal of the acylform family is shown for the different lipid classes (in supplemental) and it can be recommended that only the characteristic and main acylforms were summed for the robustness of the results. For example, PI 42:0 species is strikingly much more altered than other PI although it is likely not a major acylform of the PI family.

Response- Thank you. We have followed this advice. Figure 1c is adjusted to now show only the main 5 acylforms for each phospholipid class. The MS signals for all species detected in each acylform family are shown in supplementary files 2 and 3.

In addition the meaning of carbon number in PL or PIM annotations should be explained (total carbons or alkyl chains sum ?).

Response: Many thanks for highlighting this. We have now included the explanations (Figure 3 legend and Supplementary Figure 3 legend) and this makes the text more accessible.

2-Important accumulation of the glycerophosphothreonine head annotated in Fig 1a for the GlpQ1 mutant is not commented while it is yet unclear whether phosphatidylthreonine is even produced by mycobacteria. Figure 1a : Orange and blue color codes of are not detailed in the legend.

Response: We have now expanded our discussion of this finding, (lines 115-122). Indeed, it is intriguing, and remains unclear if phosphatidylthreonine is a minor phospholipid class in Mtb, or if it represents uptake of PThr/GroPThr from the ADC in the media (as we propose for the presence of GroPCho). Both possibilities would lead to GroPThr accumulation in the KO. Note we cannot detect GroPThr in the WT, only in the mutant.

The colour highlighting of enrichment and depletion is now detailed in the legend to Figure 1a.

3-In Figure 1c, phospholipid level changes should be shown as absolute fold change as in extended Figure 2b rather than by a heat map, which does not provide precise fold change information to the reader. Indeed, it is difficult to be convinced based on colors only that PE is unchanged and PG decreased. Red color for PE level suggests an increase in the mutant while its degradation product GroPEth is too. If fold changes are subtle, it's all the more reason to present absolute values rather than a heat map.

Response: we have followed this advice, adjusting 1c to absolute fold changes. We also now include the results from both independent experiments, which demonstrates that the changes to PIs and PGs are consistent, whereas PE, CL and PIM levels only show marginal changes which are

inconsistent and/or do not show complementation with the $\Delta glpQ1::glpQ1_Pimyc$ strain. P values (from t-tests) are now included to demonstrate which changes are statistically significant.

This contrasts the metabolic changes, where all lipid heads markedly accumulate. We interpret this as the result of homeostatic mechanisms present at the membrane, responding to the perturbation in head levels to nevertheless produce a viable and functional plasma membrane.

4-Analysis of GroPCho: this lipid polar head is detected in positive mode in Figure 2b and in negative mode in extended Figure 2a. It should be also annotated in the plot Figure 1a as it is said to be enriched in the GlpQ1 mutant.

Response

We only detect GroPCho in positive ion mode (likely due to the quaternary nitrogen). So it appears in Figure 2b and Supplementary Fig. 2a (which is also positive mode data), where indeed it is in the enriched section of the volcano plot (shown in green). Figure 1a is negative mode, where we do not detect GroPCho, so it does not appear in this plot.

There was a referencing error here- we erroneously referred to Supplementary Figure 2a as negative mode in the text when we meant to refer to figure 1a. This has been corrected (deleted line 157).

In figure 2b the screened m/z for the GroPCho head should be indicated and in 2c, the fragments in the MS2 spectrum should be interpreted with structures.

Response: Agree and these have been added to the figure.

5-Authors claimed that PC can be used by mycobacteria as a source of Pi thank to the activity of GlpQ1. Hence, PC facilitate growth in a Pi free medium without being required even for the mutant strain. The growth curves of the different strains are not very different, particularly at short time points (statistical data not available). Analysis of the GroPCho level in the WT and mutant strains grown in this PC supplemented Pi-free medium would confirm the bacillus use of exogenous PC.

Response: Thank you for this comment. To clarify on this point, we used GroPCho supplemented media in these experiments. Use of PC in the media causes turbidity, which makes OD measurements of growth impractical (we did consider using this approach as you suggest, but could not for this reason). We agree that the changes in growth between the strains are not very different, but they are:

1. Highly reproducible- the repeat experiment is shown in supplementary figure 4. The same was also seen in a very minor preliminary (i.e. third) experiment (data not shown as only included 2 strains).
2. Reflective of precedence in other organisms: *Staphylococcus aureus*, *Saccharomyces cerevisiae* and environmental pseudomonads have each been demonstrated to be able to use exogenously supplied glycerophosphodiester as sole phosphate sources, with dependence upon a either a GlpQ enzyme or an associated glycerophosphodiester membrane transporter in the same pathway [1-3].

However, ours is the first study of GlpQ in a mycobacterium, and the first to consider a link between the use of GroPCho for phosphate utilisation with the high GroPCho levels in both pulmonary surfactant within the lungs and within macrophage phagosomes.

6-What is the point of extended figure 5b? These results are not mentioned in the main text.

Response. This is an assay of PDIM levels. Loss of PDIM (phthiocerol dimycocerosates) is known to occur frequently in laboratory passaged strains of Mtb and can confound subsequent studies of virulence in animal infection (i.e. [17]). We now refer to this in the main text (lines 206-209). Thank you for highlighting the omission.

Extended figure 6c: "Total area under curve » columns are mislabeled.

Response: Thank you for highlighting this labelling error. This is now corrected.

7- The method section suffers from a lack of details in terms of data processing and bioinformatics tools, making it impossible to reproduce the analyses as it stands. To give just a few examples: Files formats, Software and their version, parameters for data processing (s/n, peak peaking parameters, RT and m/z tolerance parameters for peak alignment...), annotation method (manual or automated, ppm tolerance for annotation). In addition, raw data and extracted datasets must be made available and usable.

Response: Many thanks for this comment. We have now expanded the methods section with these details (i.e. lines 808-817 and 835-843 and 880-892). Regarding peak picking, the numerical values for signal/noise are not available, they were performed using Progenesis QI's automated method using their own noise estimation algorithm. We have now indicated the setting that we used in the methods section, which would allow users of the same software to reproduce the analysis faithfully. We have deposited the MS raw data and metadata into the Francis Crick Figshare platform, which is accessible to the public. As now detailed in the data availability statement (lines 1160-1163). The DOI will be activated prior to publication.

Articles referred to in these responses:

1. Lidbury, I., et al., *Identification of extracellular glycerophosphodiesterases in Pseudomonas and their role in soil organic phosphorus remineralisation*. Sci Rep, 2017. **7**(1): p. 2179.
2. Fisher, E., et al., *Glycerophosphocholine-dependent growth requires Gde1p (YPL110c) and Git1p in Saccharomyces cerevisiae*. J Biol Chem, 2005. **280**(43): p. 36110-7.
3. Jorge, A.M., et al., *Utilization of glycerophosphodiesterases by Staphylococcus aureus*. Mol Microbiol, 2017. **103**(2): p. 229-241.
4. Ohshima, N., et al., *Escherichia coli cytosolic glycerophosphodiester phosphodiesterase (UgpQ) requires Mg²⁺, Co²⁺, or Mn²⁺ for its enzyme activity*. J Bacteriol, 2008. **190**(4): p. 1219-23.
5. Fenn, J.S., et al., *Structural Basis of Glycerophosphodiester Recognition by the Mycobacterium tuberculosis Substrate-Binding Protein UgpB*. ACS Chem Biol, 2019. **14**(9): p. 1879-1887.
6. Jiang, D., et al., *Structural analysis of Mycobacterium tuberculosis ATP-binding cassette transporter subunit UgpB reveals specificity for glycerophosphocholine*. FEBS J, 2014. **281**(1): p. 331-41.
7. Rustad, T.R., et al., *Mapping and manipulating the Mycobacterium tuberculosis transcriptome using a transcription factor overexpression-derived regulatory network*. Genome Biol, 2014. **15**(11): p. 502.
8. Parish, T., et al., *The senX3-regX3 two-component regulatory system of Mycobacterium tuberculosis is required for virulence*. Microbiology (Reading), 2003. **149**(Pt 6): p. 1423-1435.
9. Schaeffer, M.L., et al., *The pimB gene of Mycobacterium tuberculosis encodes a mannosyltransferase involved in lipoarabinomannan biosynthesis*. J Biol Chem, 1999. **274**(44): p. 31625-31.

10. Mishra, A.K., et al., *Characterization of the Corynebacterium glutamicum deltapimB' deltamgtA double deletion mutant and the role of Mycobacterium tuberculosis orthologues Rv2188c and Rv0557 in glycolipid biosynthesis*. J Bacteriol, 2009. **191**(13): p. 4465-72.
11. Rifat, D., W.R. Bishai, and P.C. Karakousis, *Phosphate depletion: a novel trigger for Mycobacterium tuberculosis persistence*. J Infect Dis, 2009. **200**(7): p. 1126-35.
12. de Carvalho, L.P., et al., *Metabolomics of Mycobacterium tuberculosis reveals compartmentalized co-catabolism of carbon substrates*. Chem Biol, 2010. **17**(10): p. 1122-31.
13. Agapova, A., et al., *Flexible nitrogen utilisation by the metabolic generalist pathogen Mycobacterium tuberculosis*. Elife, 2019. **8**.
14. Larrouy-Maumus, G., et al., *Discovery of a glycerol 3-phosphate phosphatase reveals glycerophospholipid polar head recycling in Mycobacterium tuberculosis*. Proc Natl Acad Sci U S A, 2013. **110**(28): p. 11320-5.
15. Fisher, E., et al., *Glycerophosphocholine-dependent Growth Requires Gde1p (YPL110c) and Git1p in Saccharomyces cerevisiae**. Journal of Biological Chemistry, 2005. **280**(43): p. 36110-36117.
16. Patton-Vogt, J., *Transport and metabolism of glycerophosphodiester produced through phospholipid deacylation*. Biochim Biophys Acta, 2007. **1771**(3): p. 337-42.
17. Domenech, P. and M.B. Reed, *Rapid and spontaneous loss of phthiocerol dimycocerosate (PDIM) from Mycobacterium tuberculosis grown in vitro: implications for virulence studies*. Microbiology (Reading), 2009. **155**(Pt 11): p. 3532-3543.

Responses to reviewers

Reviewer 2

Reviewer #2 (Remarks to the Author):

1. The most important limitation of this study is the use of glycerol as a sole carbon source. This reviewer wonders about the impact of more relevant carbon sources such as cholesterol or fatty acids may have in the lipid composition of Mtb in the absence of Pi. To properly dimension the importance of the growth medium that we can mimic the infecting niche, the carbon source should be taking into consideration.

Authors Response: This is of course an important consideration and an excellent suggestion. However, the focus of this paper is on phosphate physiology rather than carbon physiology. We have previously published regarding many aspects of Mtb carbon metabolism [12], and nitrogen metabolism [13]. Furthermore, although the major carbon source in these studies is glycerol, which is widely used in studies of Mtb metabolism but is of course not relevant in infection, we also know from this study that the albumin-dextrose-catalase growth supplement used also contains multiple classes of phospholipid and fatty acids. To dissect the simultaneous contribution of available carbon, phosphate and indeed nitrogen nutrients available to Mtb upon the lipidome it displays would be a very large and complex work.

Finally, by using glycerol in these studies present in excess, whilst varying phosphate, allows us to determine that the effects seen in the phospholipids/ lipid-heads are not due to a carbon source effect.

Reviewer Response: This is not convincing. This reviewer was just suggesting checking the growth in the present of relevant carbon sources like fatty acids in controlled experiments using defined minimal medium.

Author Response:

Apologies for the misunderstanding. We did consider this aspect and we grew the strains in Sauton's media and on Lowenstein-Jenson egg white medium (i.e. fatty-acid rich)- but no differences in growth phenotype were observed- albeit both conditions still included glycerol.

5. Could authors provide some ultrastructural features of mycobacteria grown in the absence of Pi?. This would help to assess the integrity of the cells.

Authors Response: Following this suggestion with have now performed cryoEM of the WT bacteria grown in 25mM phosphate and in 0mM for 2 weeks. These results are now shown in the main manuscript in figure 4. This reveals that phosphate starved bacteria have a significant increase in the thickness of their cell envelope, as well as exhibiting a greater variance in envelope thickness, compared to bacteria grown in phosphate replete culture. This is an exciting finding, as it reflects the lipid remodelling characterised by our LC-MS studies. Please, see lines 310-317 in the new manuscript.

Reviewer Response: Authors need to provide electron micrographs of Mtb grown at 25 mM Phosphate and include it in Figure 4.

Author Response:

This is now included as suggested. Legend also adjusted.

6. There is very detailed and sophisticated mass spectrometry approach to define the lipid repertoire. However, do the authors find worthwhile to run a transcriptomic study on the glpQ1 mutant to investigate the potential compensatory mechanisms?

Authors Response: Agree this would potentially be insightful, however transcriptional analysis is not a strength of our laboratory and has not been performed in this instance. Furthermore, the regulation of enzymatic processes is not solely controlled at the level of transcription, and so the compensatory mechanisms for the alterations in phospholipid and lipid head levels in the mutant might not be decipherable from a transcriptomic approach.

Reviewer Response: This is not a convincing response. Authors speculate about the potential transcriptional regulation of this enzyme and do not open the discussion about the potential compensation of deletion of glpQ1.

Author Response:

In the manuscript we emphasise the previously published data that is that the related lipid-head transporter, UgpABCE is transcriptionally upregulated by RegX3- the master response transcription factor for responding to low phosphate. We do not make any claims about the transcriptional regulation of GlpQ1's locus itself (in Mtb).

Transcriptional analysis of *M. tuberculosis* and other mycobacterial species has been performed by multiple groups (PMID 40588129, 35980355, 33663204, 33042081, 29895636, 29326670, 28698272, 26800324, 25344463, 24722908, 23946493 and in particular 23132496), with great focus on the SenX3-RegX3, the two-component histidine kinase system that senses and transduces signals due to change in phosphate concentration. The goal of our work was to provide the much-needed biochemical changes and membrane lipid/cell envelope remodelling that accompanies physiologic concentrations of inorganic phosphate. Our lipidomic results thus represent the end-point and sum of all regulatory changes imparted by phosphate starvation.

However, as we use the transcriptional link between the transporter and low phosphate in building our hypothesis, we now add the line to the discussion:

"It would be desirable to study the GlpQ1 enzyme and the UgpABCE transporter in tandem, as the nutritional pathway could be regulated at either level, and indeed upregulation of the transporter could partially compensate for knock-out of the enzyme in the *glpQ1* mutant."

Reviewer #3 (Remarks to the Author):

The various comments have been addressed.

Minor needed edits:

Line 100: Cap missing to “higher”

Author Response:

Now corrected.

In the new figure 1 it should be mentioned in the legend that 2 independent experiments are presented.

The legend does state this for 1c. We have now added the same for 1a.

Supplementary File 2: mislabeling of PG variants m/z in the legend: “3766.5572” for the main acylform

Now corrected- 766.5572.